# New evidence of Neandertal butchery traditions through the marrow extraction in southwestern Europe (MIS 5–3)

**Delphine Vettese**[1,2,3]*, **Antony Borel**[1,4], **Ruth Blasco**[5,6], **Louis Chevillard**[1], **Trajanka Stavrova**[1], **Ursula Thun Hohenstein**[2], **Marta Arzarello**[2], **Marie-Hélène Moncel**[1], **Camille Daujeard**[1]

**1** Histoire Naturelle de l'Homme Préhistorique (HNHP, UMR 7194), Muséum national d'Histoire naturelle (MNHN), CNRS, Institut de Paléontologie Humaine, Université de Perpignan Via Domitia, Paris, France, **2** Dipartimento di Studi Umanistici, Università degli Studi di Ferrara, Ferrara, Italia, **3** Grupo de I+D+i EVOADAPTA (Evolución Humana y Adaptaciones Económicas y Ecológicas durante la Prehistoria), Dpto, Ciencias Históricas, Universidad de Cantabria, Santander, Spain, **4** Institute of Archeological Sciences, Eötvös Loránd University, Budapest, Hungary, **5** Institut Català de Paleoecologia Humana i Evolució Social (IPHES-CERCA), Campus Sescelades URV (Edifici W3), Tarragona, Spain, **6** Departament d'Història i Història de l'Art, Universitat Rovira i Virgili, Tarragona, Spain

* delphinevettese@aol.com

**Data Availability Statement:** All relevant data are within the manuscript and its Supporting Information files.

## Abstract

Long bone breakage for bone marrow recovery is a commonly observed practice in Middle Palaeolithic contexts, regardless of the climatic conditions. While lithic technology is largely used to define cultural patterns in human groups, despite dedicating research by zooarchaeologists, for now butchering techniques rarely allowed the identification of clear traditions, notably for ancient Palaeolithic periods. In this paper, we test the hypothesis of butchery traditions among Neandertal groupsusing the bone assemblages from three sites in southwestern Europe. These sites are located in southeastern France and northern Italy and are dated to the Late Middle Palaeolithic: Abri du Maras (Marine Isotopic Stages (MIS) 4–3, Ardèche), Saint-Marcel (MIS 3, Ardèche), and Riparo Tagliente (MIS 4–3, Verona). The detection of culturally-induced patterns of bone breakage involves differentiating them from intuitively generated patterns. To tackle this issue, we used a zooarchaeological approach focusing on the percussion marks produced during the bone breakage process. Statistical analyses as the chi-square test of independence were employed to verify if percussion mark locations were randomly distributed, and if these distributions were different from the intuitive ones. For femurs and humeri, our results demonstrate that Neandertal groups occupying the Abri du Maras (levels 4.1 and 4.2) and the Saint-Marcel Cave (levels g and h) sites in France applied butchery traditions to recover yellow marrow. However, the traditions developed at each site were different. On the contrary, in Riparo Tagliente, in Italy, several groups or individuals of a same group did not share the same butchery traditions over time. Regarding the Abri du Maras and Saint Marcel Cave assemblages, our research demonstrates that Neandertal groups applied intense standardized bone breakage, far from the intuitive practice observed experimentally and related to bone density and/or skeletal morphology. These standardized patterns, which are systematic and counter-intuitive, can be interpreted as culturally induced for the Abri du Maras and Saint Marcel Cave. The diversity of Neandertal

**Funding:** RB:Ramón y Cajal research contract by the Ministry of Economy and Competitiveness (RYC2019-026386-I) and AEI/FEDER, EUproject PID2019-104949GB-I00, and the Generalitat de Catalunya projects 2017 SGR 836 and CLT009/18/00055; CD, DV, TS and LC was supported by the Fondation Nestlé France (SJ 671–16) (https://fondation.nestle.fr/); DV is supported by the Centre d'Information des Viandes – Viande, sciences et société (SJ 334–17); AB, CD, DV, LC, MHM and TS are supported by the Muséum national d'Histoire naturelle. The funders had no role in study design, data collection and analysis, decision to publish, or preparation of the manuscript.

**Competing interests:** The authors have declared that no competing interests exist.

traditions should be considered by taking into account the butchery, in particular the practice of bone marrow extraction, and not only technological behaviours and types of tool kits.

## Introduction

Over the last two decades, the diversity and flexibility of the Neandertal diet have been widely demonstrated [1–9]. The ability to use and to cook all the available resources in the environment suggests a combination of complex subsistence behaviours and opportunism [10–16]. Isotopic analyses show a dietary strategy based primarily on animal protein[17–19]. In addition to the consumption of meat from large and small game, bone assemblages in Neandertal sites indicate systematic fat consumption [9, 20–24]. For example, red and yellow bone marrow was an important food resource, especially in dry and cold environments [25–28]. Given the large accumulations of broken bones by Neandertal in archaeological sites, focusing on the modalities of bone fracturing seems to be an innovative way of tracking butchery traditions among Neandertal groups [1, 29–35].

For decades, identification and analyses of Neandertal traditions through time, in specific site and comparing sites with others were based on studies of lithic assemblages, which were assumed to record the characteristics of a group with the inter-generational transmission of knowledge [36–38]. Stone tools are even used as markers of a specific group or regional traditions. More and more, lithic analyses focus on the hypothesis that stone tool corpuses may indicate both activities and traditions. Regarding the analyses of faunal assemblages, the identification of traditions is mostly tested through studies of hunting strategies [33, 39–42] while butchery techniques are rarely considered.

In this paper, we describe standardized and counter-intuitive patterns of breaking bones. These patterns are consistent with butchery traditions shared and transmitted within a same Neandertal group. Intuitiveness is the immediate intuition of a non-trained butcher to break a bone. It could be influenced by anatomical constraints as the morphology, the thickness of cortical bone, the tissues compacta or spongiosa [43, 44]. Some experiments focusing on the intuitive way to extract marrow highlighted intuitive patterns of percussion mark distribution [43, 45, 46]. Thus, the development of specific butchery skills, which could be applied to Neandertals who regularly broke long bones to extract yellow marrow, is a valuable hypothesis [47]. Hence, transmitted skills include habits enhanced by experience and/or group traditions. Butchery traditions imply know-how (sequences of gestures), dedication (time involved in the activity) and skill (ability to reproduce gestures and to correct them) [48].

The transmission of butchery knowledge from one generation to another is essential in our definition of tradition. A majority of the Middle Palaeolithic sites and levels is a palimpsest of several occupations, which means one or more groups could have successively occupied a level. The identification of one butchery tradition within a level means the identification of one group, with members of different generations or over several generations, returned to the site multiple times. This hypothesis is based on ichnology or genetic studies, which have focused on the composition of the Neandertal group [49–51]. For example, the Neandertal group of le Rozel was mainly composed of children and teenagers/youngsters. This group is composed of a relatively reduced number of individuals, who repeatedly occupied a level. Blasco et al. [43] tested, for the first time, the hypothesis of butchery practices at two Middle Palaeolithic Spanish sites (MIS 5–9): Bolomor (levels IV, IX, XIIa and XIIc) and Gran Dolina (TD10-1 [faunal sample from field-work seasons 2000–2001]). They highlighted the presence

of systematic and counter-intuitive percussion mark patterns in level IV of Bolomor, indicating specific butchery traditions. On the contrary, in the earlier TD10-1 level of Gran Dolina, they could not demonstrate cultural know-how but brought to light a probable palimpsest combining various practices. After their innovative paper, a few other analyses tried to demonstrate the existence of butchery traditions to recover yellow marrow in the southern Mediterranean area [29, 52].

Unlike percussion marks, cutmarks appeared less successful in identifying standardized butchery practices, mainly because of the role of the butcher's dexterity, animal morphology and size, and raw materials and tools used, all having a deep influence on the frequency and location of cutting incisions [53, 54]. Indeed, cutmarks made during carcass processing may be scarce for experimented butchers and are mainly accidental because most of the cutting focus on soft tissues and not directly on bones. However, the direct percussion of long bones is an action on bone when almost all the soft tissue are already removed [27, 29, 43, 55, 56]. Besides, percussion marks are part of a complex "*chaîne opératoire*" where the bone can be considered as a raw material both for alimentary purposes and for tool production, i.e., bone retouchers or hammers for Neandertals [57–59].

From the point of view of cognitive capacities, butchery traditions should be embedded within the socio-cultural practices of Neandertal groups, in the same way as hunting strategies and the cultural traditions of lithic tools. To test this hypothesis, we focused on three late Neandertal sites with anthropogenic faunal accumulations: Abri du Maras and Saint-Marcel Cave in southeastern France, and Riparo Tagliente in northeastern Italy. In order to monitor possible behavioural changes in butchery practices over time, we selected two layers from each site, all dated between Marine Isotopic Stages (MIS) 4 and 3 (Fig 1). We only focused on percussion mark distribution produced by yellow marrow recovery, after the evaluation of the impact of taphonomic modifications on the archaeological assemblages.

The Middle Palaeolithic faunal assemblages of Abri du Maras, Saint-Marcel Cave and Riparo Tagliente are appropriate because they are almost exclusively anthropogenic accumulations, with limited carnivore marks on bones (e.g., [60–63]). The faunal remains are well preserved and a complete butchery "*chaine opératoire*" was conducted on site. Moreover, regarding each assemblage, all the bones were systematically recorded, including the smallest bone fragments. For almost all of them, the faunal spectrum is practically monospecific (except

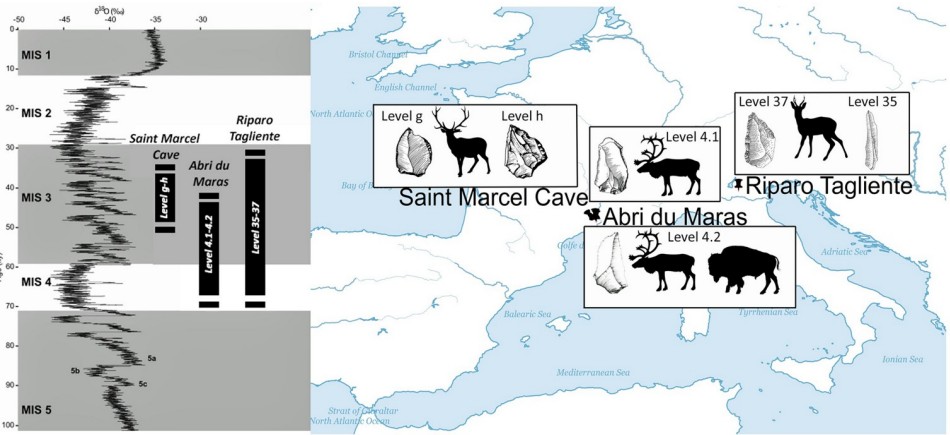

**Fig 1. Location of the studied sites with the hunted species and an illustration of the lithic culture.** Drawing by M. Arzarello, R. Gilles and A. Theodoropoulou. The map was created using ArcGIS (ArcMap 10.4), and uses Natural Earth vector map data, (https://www.naturalearthdata.com/downloads/).

the level 4.2 of Abri du Maras), with the predation of cervids (roe deer, red deer and reindeer), which is propitious to the comparison of hunting strategies. Our results are then compared with previous data published by Blasco et al. [23] and with our own experimental data *e.g.*: [45].

## Archaeological background

### Abri du Maras levels 4.1 and 4.2

The site of Abri du Maras is located on the right bank of the Middle Rhône Valley, in a small dry valley of the Ardèche River, 70 m above the river. It constitutes the remains of a huge, former south-east facing rock shelter [30, 37, 60, 64].

The first archaeological investigations revealed Middle Palaeolithic deposits, with Levallois laminar debitage at the top of the sequence [64–67]. The new excavation (since 2006), directed by M.-H. Moncel, only concerned the middle and lower parts of the sequence. Six layers were described. In this paper, we are focusing on the layer 4 consisting of a silt and sandy-silty sedimentary accumulation (0.5 to 1 m thick), with two main phases of human occupations (4.1 and 4.2), with traces of combustion and occasional diffuse ash lenses. Layer 4 was excavated over an area of more than 50 m$^2$ [68]. There was no significant orientation of the material suggesting that no major disturbance of the archaeological remains occurred [69].

U/Th dating applied to bones from the top of unit 5/bottom of layer 4 yielded ages of 72 ± 3 ky, 87 ± 5 ky, 89 ± 4 ky, and 91 ± 4 ky [67, 70]. New preliminary ESR-U/Th dating of ungulate teeth confirmed the chronology of the underlying layer 5 (90 ± 9 ky) but gave more recent ages for layer 4, attributed to MIS 3. Level 4.1 (upper part of layer 4) dates to between 40 ± 3 ky and 46 ± 3 ky (n = 2) (also MIS 3), while samples from level 4.2 provide ages ranging from 42 ± 3 ky to 55 ± 2 ky (n = 3) (MIS 3) [71].

The faunal spectrum of the unit 4 is composed, in order of abundance, of *Rangifer tarandus*, *Equus ferus* cf. *germanicus*, *Cervus elaphus*, *Bison priscus*, *Capra ibex*, *Equus hydruntinus* and *Megaloceros giganteus* [30, 37, 61, 62]. Carnivore remains were lacking. Contrary to level 4.1, where reindeers largely dominate with almost 90% of the NISP, level 4.2 does not display such a mono-specific spectrum [61]. Large ungulates, as horses and bisons, as well as red deers, have a greater frequency compared to reindeers.

Lithic analysis revealed that the technical strategies applied mainly to flint and secondarily to other stones (quartz, quartzite and granite) into local and semi-local perimeters around the site [68]. The main methods of core exploitation are Levallois, discoidal and expedient, generally on cortical flint cores on flakes. Flint flakes, secondary blades and points are the main components. The largest flint flakes, Levallois blades and points were brought to the site from up to 20–30 km away (north and south) from the site. Flake-tools are very rare (3% for level 4.1 and 4.8% for level 4.2), and the retouch never modified the shape of the pieces, except for one and two Quina scrapers made on a thick and wide flint flake. In level 4.1, 32 pebbles and in level 4.2, 21 pebbles, sometimes broken or with a single removal, were introduced, possibly as hammerstones, although they bear few percussion marks. These quartz, quartzite, limestone, basalt, granite and rarely flint pebbles were collected in the surrounding rivers. They measure mainly between 60 and 120 mm and are round or oval. The lithic analyses showed the anticipation of domestic needs for butchery, plant and woodworking [72–74]. In this site, Neandertals exploited a wide range of resources including large mammals, fishes, rabbits, mushrooms, plants and wood [6, 72, 74]. Evidence of impact fractures suggests that some points could have been used as projectile tips. They were either brought to the site as points or prepared on the site [72].

## Saint Marcel Cave layers g and h

The site of Saint-Marcel is located at the end of the Ardèche Gorges, not far from the Abri du Maras, on the north side of the gorge, approximately 200 m inland from the Ardèche River. The deposits lie under the cave overhang, in a very large, sub-horizontal, south-facing section, 12.5 m wide, 7 m deep and 3 m high. Excavations of the Mousterian layers were carried out by R. Gilles from 1974 to 1988, during which a surface area of 30 m² was exposed, reaching a depth of nearly 7 m. Two main stratigraphic sedimentary units were defined. The upper one contains the twelve Mousterian occupation phases, the lower one is sterile [64, 75].

The lower layers (k to u) were attributed to MIS 5 based on the sedimentological analysis [64]. The palaeontological analysis of the faunal remains clearly set layer u apart from all the others, given the abundance of fallow deer in this layer, with *Hemitragus cedrensis*, as well as a similar red deer to that one found at the site of Les Cèdres (Var, France). These taxa point to an Eemian occupation for layer u [76–78]. Palaeontological, environmental and sedimentological analyses attribute Unit 7 (layers j to g) to a warm episode, either MIS 3 or the end of MIS 5 [62, 64, 79–82]. There seemed to be no hiatus in sedimentation between layers j and g, nor between g and f [62, 64, 80]. $^{14}$C results on bones from layer f [83] were statistically identical to each other: 37 850 ± 550 BP and 37 850 ± 600 BP. The third, 41 300 ± 1700 BP, overlaps them at two standard deviations. Thus, the chronological attribution of Unit 7 seems to be rather MIS 3

In all of the upper layers at Saint-Marcel, red deer represents more than 80% of the number of identified specimens (NISP), with the exception of the level u where the fallow deer is dominant. The faunal spectrum of the levels g and h is composed of, in order of abundance: *Cervus elaphus*, *Capreolus*, *Capra caucasica*, *Dama* sp., *Sus scrofa* and *Equus* sp. [62, 64]. Neandertals practised selective hunting of prime-aged adult red deer [62, 79, 80]. Carcass butchery was particularly intensive. Cutmarks and percussion marks were recorded on just over half the identified specimens in the faunal assemblage and more than 70% bore evidence of green-bone fractures. Topographical and zooarchaeological data showed that all the layers were associated with long-term occupation. In addition, the high proportion of bone retouchers (n = 303, representing nearly 6% of the total number of remains with a size superior to 5 cm (NR)) makes this site distinctive.

The lithic assemblages show marked homogeneity throughout the stratigraphic sequence. Flint, which was collected in various forms (as pebbles, nodules and tablets) was the main raw material used, and was procured from a variety of sectors (in the Barremian-Bedoulian from the plateau to the north, from the Orgnac plateau to the south and in the Rhone Valley to the east) [75, 84]. The richest phase of occupation corresponds to the layers i and h, in the middle part of the sequence. Most of the flint processing probably took place at the site; the entire *chaîne opératoire* is represented. The lithic assemblage from all the layers consisted mainly of debitage products, especially flakes. These were small in size, ranging from 30 to 50 mm long, on average. They could mostly be attributed to discoid debitage (centripetal, unipolar or bipolar method), carried out especially on the ventral surface of flakes. In some cases, they strike along a third plane, resulting in the extraction of thick flakes with a triangular cross-section ("crested flake") [75]. Very few retouched tools were found at this site (approximately 5%), consisting primarily of scrapers, in addition to some convergent pieces. Retouch was light and did not modify the blank, although some thinning is present.

For the other raw materials, the number of lithic objects is low and consists of some entire pebbles and above all pebble fragments/flakes in quartz, limestone and quartzite. Little evidence of the role of these raw materials was identified with remains of hammerstones or introduction of some flakes.

## Riparo Tagliente layers 35 and 37

Riparo Tagliente is a rock shelter located in the village of Stallavena di Grezzana (Verona, N-E Italy). It is located at the bottom of a rock wall, formed by limestone, on the west slope of Valpantena, one of the main valley-bottom of the pre-Alpine massif of Monti Lessini at an altitude around 250 m asl [85–87].

The site was discovered in 1958 by F. Tagliente and excavated by the Museo di Storia Naturale di Verona from 1962 to 1964 [88–91]. In 1967, the University of Ferrara taken over the excavation of the site and pursue it until now. The systematic excavations provided an important stratigraphic series of deposits attributed to the Upper Pleistocene, within a high anthropogenic occupation [92].

Until the mid-seventies research was focused mostly on the excavation of a long trench running transversally to the rock-shelter between two test-pits: one located in the most internal area (southern sector, cf. Fig 1 in [93]) and another one situated in the most external area (northern sector). Two main archaeological units separated by a river escarpment were excavated. The lowermost unit, characterized by Mousterian and Aurignacian lithic industries, was occupied during the MIS 4–3, and the uppermost unit, which provided an Epigravettian industry, was occupied during the Late Glacial. The layer 35 and 37 belonged to the upper layers of the Mousterian deposits, characterised by loess sediment. No radiometric dating was performed yet, the dating estimation of the lower unit was based on the analyses of the lithic and faunal assemblages [86, 94].

The ungulates dominated the faunal spectrum of the two studied layers. *Capreolus* is the most represented species, *Cervus elaphus*, *Capra ibex* and *Rupicapra* are following regarding the NISP. Among the few remains of carnivores, *Canis lupus* and *Ursus* sp. were the most represented. Few remains of *Marmota* have been identified, including a mandible with cutmarks [1, 95] and few non-identified remains of Aves and only one of *Lepus* sp. The presence of foetal or neonatal cervid bones suggests an occupation of the rock shelter mainly during the spring-time [63, 96]. Long bones constitute most of the faunal assemblage of the layers 35 and 37. Some isolated teeth, remains of axial skeleton, few tarsals and carpals and numerous phalanxes were identified.

Within the layer 37, two teeth attributed to *Homo neanderthalensis* were recovered [93, 97].

In layers 35 and 37, the use of different reduction methods on local raw materials (chert) characterized the Mousterian sequence. These cherts were collected in the surrounding of the site mostly in secondary position. The opportunistic method was the best represented (c.f. *Système par Surface de Débitage Alternée* [98]). Lithic analyses also highlight the presence of discoid and Levallois methods with the lineal and recurrent modalities, in the upper layers the unipolar recurrent became predominant. The retouched blanks are very numerous and mainly represented by side scrapers and denticulates [99–101]. The presence of a volumetric laminar debitage starting from layer 37 represented one of the main particularities of the lithic assemblage [100].

Very few cobbles have been found in the Mousterian layers and they all show percussion marks; they have consequently been interpreted as hammerstones. All pebbles are mostly spherical and have a diameter between 60 and 85 mm. They have been collected in the surrounding of the site and are mainly in quartzite (only one is on limestone). In the layer 35 no cobble has been found and in the layer 37, within the innermost part of shelter, only 3 pebbles have been found.

## Material and methods

### Samples selection

The Number of Remains (NR); the Number of anatomically Identified Specimens (NISPa); Number of taxonomically Identified Specimens (NISPt); the Minimum Number of Elements

(MNE); the Minimum Number of Individuals (MNI) and the Minimum Animal Unit (MAU) were the quantitative indices used in this work [27, 55, 102]. The MNE was calculated based on the count of the preserved areas (cf. Fig 2 in [29]) [43]. For the taxonomical and anatomical determination, we used both anatomical atlases and comparative collections located at the University of Ferrara and at the IPH (Institut de Paléontologie Humaine, Paris France), e.g. [103–105]. We based our identification on palaeontological and previous zooarchaeological studies [61, 62, 106]. Each identified faunal remain was grouped in one of three main size categories: Large-sized ungulates (LUNG), Medium-sized ungulates (MUNG) and Small-sized ungulates (SUNG) [41, 107] (S1 Table in S1 File).

We followed the methodology established by Vettese et al. [108] to analyze percussion marks distribution. Our work is a continuation of this paper, focusing on the comparison of percussion marks location on long bone shafts. For this reason, only long bone remains identified at least anatomically are taken into account, in order to accurately place percussion traces on bone. In our work, we distinguished long bone size categories for analytic purposes, i.e., SUNG, MUNG and LUNG. Within a given class size category, only samples with an MNE greater than 100 have been selected.

For this study, we considered the faunal assemblages from Abri du Maras (layers 4.1 and 4.2), Saint Marcel Cave (layers g and h) and Riparo Tagliente (layers 35 and 37). For the purposes of an intra-site comparison, we selected two relatively stratigraphically close layers from each site, coming from the same depositional unit. The layers chosen were highly anthropized, with butchery marks and very limited carnivore modifications. The whole material represented more than 2,180 remains. The archaeological material of the Saint Marcel Cave is preserved in the *Cité de la Préhistoire* of Orgnac (France), that of the Abri du Maras is temporary deposited in the *Institut de Paléontologie Humaine* of Paris (France) and will be transferred at in the *Cité de la Préhistoire* of Orgnac (France) and that of Riparo Tagliente is in the University of Ferrara (Italy). All necessary permits were obtained for the described study, which complied with all relevant regulations.

The results concerning the percussion marks in layer 4.1 of the Abri du Maras were previously presented in [29]. For comparative purposes, we included the results relating to fragmentation, taphonomy and percussion mark location in the present analyses. Regarding the faunal spectrum of layer 4.2, we divided this assemblage into two different samples named 4.2LUNG and 4.2MUNG. They constituted respectively 39% and 51% of the NISPt. The sample from this layer provided 570 remains, including 203 4.2LUNG and 367 4.2MUNG.

The studied material from Saint Marcel Cave comes from layers g and h. These two layers were dominated by the MUNG, in particular by the red deer (layer g: $NISPt_{MUNG}$ 91% of the $NISPt_{Total}$; layer g: $NISPt_{MUNG}$ 87% of the $NISPt_{Total}$) [62]. Our analyses were carried out on 258 remains from layer g and 303 long bone remains from layer h.

The analysed material from Riparo Tagliente comes from layers 35 and 37. Most of the sample consists of roe deer for both layers and is completed by ibex and chamois remains (layer 35: $NISPt_{SUNG}$ 90% of the $NISPt_{Total}$; layer 37: $NISPt_{SUNG}$ 44% of the $NISPt_{Total}$). SUNG long bone remains dominated the faunal assemblages of these layers. We took into account 176 SUNG remains from layer 35 and 138 remains from layer 37.

## Taphonomy

The cortical surface of each remain was observed with the naked eye and a 15-20x lens, under low-angled light. When necessary, a stereomicroscope (Leica S8 APO 10-80x/ Leica MZ6 10-40x) was used for the identification and recording of bone surface modifications. Different taphonomic modifications (edaphic and biologic) were recorded based on the criteria

identified in the scientific literature e.g., [27, 55, 109–114]. In order to identify the impact of taphonomic alterations, we recorded which part of the cortical surface was altered by each physico-chemical taphonomic modification, following [62]. More specifically, for each alteration modifying the surface of the bone fragment, we attributed a scale code from 0 to 3. The code was 3 when the modification covered the entire bone surface. The code was 2 if more than half of the entire bone surface was modified. If less than half the surface was affected, the code was 1. Finally, if no modification was observable to the naked eye, the code was 0. Taking into account all the modifications, we noted that some/several cortical surfaces could be completely dissolved or concreted until they were unreadable. Unreadable remains were also counted for each assemblage. We compared the different surface modifications for the studied sites and layers.

We paid special attention to anthropogenic modifications and in particular to percussion marks. Burnt damage was also recorded [115]. We divided the cutmarks into two categories, according to the criteria established by [116]; incisions and scraping marks. We excluded from the percentages of the cutmarks analysis all the remains with unreadable bone surfaces. In the studied faunal assemblages, bone retouchers were recorded, and again, the unreadable remains were excluded from the percentages. They were identified based on recent works setting out new criteria [59, 117–119].

The long bone fragmentation of our assemblages was compared in terms of length: these classes are 0–25 mm, 26–50 mm, 51–75 mm, 75–100 mm, 101–125 mm, >125 mm, bone part (complete epiphyses) and whole bones [120]. We measured the three dimensions (length, width, thickness) for each element. According to their size and circumference, we categorized the shaft fragments [121]: $l1 < 1/4 \leq l2 < 1/2 \leq l3 < 3/4 \leq l4 = 1$ (complete) and $0 < C1 < 0.5 \leq C2 < 1$; $C3 = 1$ (complete). Based on fracture features, we distinguished between dry bone fractures, recent breakage, gelifraction and green bone fractures [120, 121].

The percussion marks were recorded following the terminology of [108]: crushing marks, notches, adhering flakes, pits and grooves. In this work, we did not include flakes because it was virtually impossible to locate them precisely on the long bone shaft. In order to identify preferentially impacted areas, each identified fragment was positioned in an area according to its long bone portion and side (cf. Fig 2 in [29]). Some bone fragments could be identified based on an anatomical particularity, such as the metatarsal gutter, but it was not possible to locate precise areas. In such cases, they were excluded from percussion mark distribution analyses.

The number of areas preserved by element varies according to the analysed faunal assemblages. We weighted the number of areas with percussion marks with the number of areas preserved for each area. The percussion mark ratio is calculated by: Number of areas with percussion marks / Number of preserved areas.

### Experimental data

For the purposes of comparison, we included experimental data from three experiments [43–45, 122]. The main aim of these experiments was to record percussion mark location when novice experimenters broke bones to extract the yellow marrow. The analysis of percussion mark distribution along the long bone shaft provided data on intuitive patterns. These three experiments enable us to compare two different species ([43] and [122] for cow, and [44] for red deer) and the technique used to break the bones. Indeed, some experimenters used the hammerstone on anvil technique [43–45, 122]; while the rest of them used the batting technique [43, 122]. The analyses conducted to record percussion mark distribution are similar to those applied to the archaeological samples, presented above.

## Data analysis

We used an alpha of 0.05 for all the tests. Spearman's rho was used to test the correlation between long bone density indices and percussion marks frequency by portion to test whether the densest parts or the least dense zones were the most impacted. Bone density indices were estimated by [55] and [123] LUNG for *Equus*, MUNG for *Cervus* and SUNG for *Ovis*. We also employed this bivariate test to cross the %MAU and the medullary cavity volume of ungulate long bones to explore human carcass transport strategies [124]. Finally, we tested the correlation between the amounts of percussion marks per element with its NISPa to determine whether the number of identified fragments affected the number of percussion marks recorded per element. The correlation coefficients and tests were computed using R software [125].

The chi-square test of independence was employed to verify if percussion mark locations were randomly distributed along the diaphysis or not. The chi-square test of conformity was used to verify if the observed distributions are similar to intuitive butchery's distribution.

Multiple Correspondence Analysis (MCA) was used to investigate the relationship between the impacted and non-impacted bone portions (the three shaft portions) and bone sides and to examine whether butchery patterns could be identified for specific sites and/or levels and bone elements. The site, the level and the bone elements are displayed as illustrative variables. MCA was computed using the FactoMineR [126] library in R. When Cochran's rules were not respected, we used Fisher exact tests with R to test the independence between the locations of impacts on the long bone shafts of each assemblage according to the different long bones. When necessary, pair wise multiple testing was performed. In such cases, the Holm correction of multiple testing was applied to the p-values.

In the MCA, our samples were compared with archaeological samples from Bolomor (IV, XVIIa, XVIIc) and Gran Dolina (TD-10-1 [upper part]) and with four experimental series [43–45, 122]). The comparison of the results of [43] with two Middle Palaeolithic Spanish sites (MIS 5–9): Bolomor (levels IV, IX, XIIa and XIIc) and Gran Dolina (TD10-1), enables us to test the use of a standardized method for recovering marrow. Their results highlighted the presence of percussion mark patterns in Bolomor (level IV), indicating specific butchery traditions. On the contrary, in level 10–1 of Gran Dolina, the study did not demonstrate the presence of a tradition. We compared all the long bones. Metapodials were absent in the experimental samples, therefore, the comparison of these elements was only carried out for the archaeological samples.

## Results

### Taphonomic preservation of the bone samples

**Bone surface preservation.** The taphonomic alterations affect our various bone assemblages differently (Table 1), but they are quite similar within each sequence. The vast majority of the long bones of Saint Marcel and Riparo Tagliente have very-well preserved bone surfaces, while one-third of the bone surfaces are unreadable at Abri du Maras in both layers. This bad bone surface preservation derives mainly from root-etching dissolution, and in some cases prevents the identification of cutmarks, bone retouchers and percussion pits. Notches are not usually affected by surface illegibility. Bone cracking is recorded on all the assemblages. Concretions are limited on bone surfaces, in particular at Abri du Maras layer 4.2 and Riparo Tagliente.

Concerning animal-induced modifications, digested elements and carnivore or rodent tooth marks are either scarce or inexistent in all the sites. The site of Riparo Tagliente is the most affected, with only three elements bearing carnivore marks [29, 62, 63].

**Table 1. Bone surface alterations (NR and τ) with for each of the stages (3 to 0) of surface modifications for each level of each site, Saint Marcel Cave layer g and h; Riparo Tagliente layer 35 and 37); Abri du Maras layer 4.1, 4.2 LUNG and 4.2 MUNG.**

| | Sites | Saint Marcel Cave | | | | Riparo Tagliente | | | | Abri du Maras | | | | | |
|---|---|---|---|---|---|---|---|---|---|---|---|---|---|---|---|
| | layer | g | % | h | % | 35 | % | 37 | % | 4.1 MUNG | % | 4.2 LUNG | % | 4.2 MUNG | % |
| Cracking | f0 | 96 | 37.21 | 102 | 33.66 | 88 | 63.78 | 104 | 59.09 | 452 | 60.92 | 20 | 9.84 | 75 | 20.44 |
| | f1 | 107 | 41.47 | 105 | 34.65 | 31 | 22.46 | 53 | 30.11 | 247 | 33.29 | 59 | 29.05 | 160 | 43.60 |
| | f2 | 32 | 12.41 | 69 | 22.77 | 14 | 10.14 | 12 | 6.82 | 40 | 5.39 | 65 | 32.01 | 93 | 25.34 |
| | f3 | 023 | 8.91 | 27 | 8.91 | 5 | 3.62 | 7 | 3.98 | 3 | 0.4 | 59 | 29.1 | 39 | 10.62 |
| Concretion | c0 | 226 | 87.60 | 241 | 79.54 | 112 | 81.16 | 110 | 62.5 | 646 | 87.06 | 17 | 8.37 | 63 | 17.17 |
| | c1 | 30 | 11.62 | 47 | 15.51 | 18 | 13.04 | 56 | 31.82 | 85 | 11.46 | 157 | 77.34 | 281 | 76.56 |
| | c2 | 2 | 0.78 | 15 | 4.95 | 6 | 4.35 | 8 | 4.54 | 11 | 1.48 | 29 | 14.29 | 23 | 6.27 |
| | c3 | - | - | - | - | 2 | 1.45 | 2 | 1.14 | - | - | - | - | - | - |
| Dissolution | s0 | 185 | 71.71 | 222 | 73.27 | 26 | 18.84 | 59 | 33.52 | 18 | 2.43 | 2 | 0.99 | 6 | 1.63 |
| | s1 | 58 | 22.48 | 68 | 22.44 | 110 | 79.71 | 114 | 64.77 | 281 | 37.87 | 26 | 12.81 | 93 | 25.34 |
| | s2 | 15 | 5.81 | 13 | 4.29 | 2 | 1.45 | 3 | 1.71 | 267 | 35.98 | 75 | 36.94 | 144 | 39.24 |
| | s3 | - | - | - | - | - | - | - | - | 176 | 23.72 | 100 | 49.26 | 124 | 33.79 |
| Blunting | b0 | 208 | 80.62 | 189 | 62.38 | 129 | 93.48 | 167 | 94.89 | 688 | 92.73 | 98 | 48.28 | 232 | 63.21 |
| | b1 | 44 | 17.05 | 103 | 33.99 | 9 | 6.52 | 9 | 5.11 | 51 | 6.87 | 79 | 38.92 | 124 | 33.79 |
| | b2 | 6 | 2.33 | 11 | 3.63 | - | - | - | - | 3 | 0.4 | 24 | 11.8 | 11 | 3.00 |
| | b3 | - | - | - | - | - | - | - | - | - | - | 2 | 0.98 | - | - |
| Desquamation | d0 | 207 | 80.22 | 251 | 82.84 | 51 | 36.96 | 73 | 41.48 | 206 | 27.76 | 40 | 19.70 | 91 | 24.80 |
| | d1 | 35 | 13.57 | 44 | 14.52 | 69 | 50 | 95 | 53.98 | 170 | 22.92 | 108 | 53.2 | 151 | 41.14 |
| | d2 | 14 | 5.43 | 8 | 2.64 | 18 | 13.04 | 7 | 3.98 | 200 | 26.95 | 45 | 22.17 | 100 | 27.25 |
| | d3 | 2 | 0.78 | - | - | 0 | - | 1 | 0.57 | 166 | 22.37 | 10 | 4.93 | 25 | 6.81 |
| Root-etching | r0 | - | - | - | - | 35 | 25.36 | 81 | 46.02 | 29 | 3.91 | 4 | 1.97 | 13 | 3.54 |
| | r1 | - | - | - | - | 102 | 73.92 | 94 | 53.41 | 246 | 33.15 | 33 | 16.26 | 96 | 26.16 |
| | r2 | - | - | - | - | 1 | 0.72 | 1 | 0.57 | 270 | 36.39 | 80 | 39.41 | 160 | 43.60 |
| | r3 | - | - | - | - | - | - | - | - | 197 | 26.55 | 86 | 42.36 | 98 | 26.70 |
| Black colouring | | - | - | - | - | 116 | 84.06 | 159 | 90.34 | 590 | 79.51 | 183 | 90.15 | 331 | 90.19 |
| Orange colouring | | - | - | - | - | 80 | 57.97 | 119 | 67.61 | 365 | 49.19 | 82 | 40.39 | 139 | 37.87 |
| Unreadable | | 2 | 0.78 | 1 | 0.33 | 5 | 3.62 | 4 | 2.27 | 180 | 24.26 | 87 | 42.86 | 123 | 33.51 |
| Total | | 258 | 100 | 303 | 100 | 138 | 100 | 176 | 100 | 742 | 100 | 203 | 100 | 367 | 100 |

**Long bone element distribution and differential bone preservation.** The frequencies of the different long bones vary according to the layers and ungulate class-size. The %MAU of the long bones of Riparo Tagliente is the highest since almost all the elements have a ratio superior to 50% (Table 2 and S1 Fig in S1 File). Tibias are the most abundant long bones for all the assemblages, except in layer 4.1 (Abri du Maras), where metatarsals are the most frequent (Fig 2). Radio-ulnas, except for AM-4.2LUNG assemblage, are also numerous.

The %MAU of each long bone and the marrow cavity volume (ml) are not significantly correlated (Spearman correlation: df = 5; p-value > 0.05). On the contrary, the %MAU and the bone density are significantly and positively correlated for: Abri du Maras, assemblages 4.1 and 4.2MUNG; Saint-Marcel Cave g and h (Spearman correlation: df = 25; p-value < 0.01; $r_s$ < 0.01) and Riparo Tagliente layer 35 (Spearman correlation: df = 25; p-value = 0.01; $r_s$ = 0.45). No significant correlation is noted for layer 37 of Riparo Tagliente and the 4.2LUNG assemblage of Abri du Maras (Spearman correlation: df = 25; p-value > 0.05).

**Long bone breakage.** All the long bones are highly fragmented (S4-S9 Figs in S1 File). No complete long bones were found at Riparo Tagliente and Saint Marcel. At Abri du Maras, we only identified metapodials: one complete reindeer metacarpal in layer 4.2, and three reindeer

**Table 2.  Number of anatomically Identified Specimen (NISPa), Left (L) and Right (R) sides, Number of Remains belonging to juveniles (juv), Minimum Number of Element (MNE), Minimum Animal Unit (MAU), maximal length (Max), minimal length (Min), mean size of long bones (Mean) and standard deviation (SD).**

| Sites | Element | NISPa | % NISPa | Right | Left | juv | NME | % NME | MAU | % MAU | Max (mm) | Min (mm) | Mean (mm) | SD |
|---|---|---|---|---|---|---|---|---|---|---|---|---|---|---|
| Saint-Marcel Layer g MUNG | Humerus | 18 | 7 | 12 | 6 | 2 | 9 | 8 | 4.5 | 24 | 119 | 39 | 69.89 | 20.06 |
| | Radio-ulna | 41 | 16 | 21 | 14 | 1 | 24 | 20 | 12 | 65 | 157 | 30 | 88.68 | 28.59 |
| | Metacarpal | 31 | 12 | 7 | 9 | 0 | 13 | 11 | 6.5 | 35 | 143 | 33 | 71.81 | 29.69 |
| | Femur | 32 | 12 | 12 | 10 | 2 | 16 | 13 | 8 | 43 | 149 | 36 | 88.88 | 28.07 |
| | Tibia | 64 | 25 | 18 | 35 | 0 | 37 | 31 | 18.5 | 100 | 181 | 38 | 92.82 | 29.89 |
| | Metatarsal | 47 | 18 | 6 | 12 | 1 | 18 | 15 | 9 | 49 | 156 | 32 | 76.71 | 27.52 |
| | Metapodial | 25 | 10 | 3 | 3 | 1 | 3 | 3 | 1.5 | 8 | 159 | 9 | 77.56 | 28.82 |
| | **Total** | **258** | | **79** | **89** | **7** | **120** | | **60** | | **181** | **9** | **83.09** | **29.34** |
| Saint-Marcel Layer h MUNG | Humerus | 30 | 10 | 12 | 14 | 1 | 17 | 13 | 8.5 | 52 | 138 | 39 | 73.06 | 23 |
| | Radio-ulna | 43 | 14 | 19 | 19 | 1 | 17 | 13 | 8.5 | 52 | 137 | 34 | 73.58 | 25.89 |
| | Metacarpal | 45 | 15 | 7 | 7 | 2 | 13 | 10 | 6.5 | 39 | 128 | 30 | 72.02 | 22.42 |
| | Femur | 36 | 12 | 14 | 14 | 0 | 18 | 14 | 9 | 55 | 154 | 29 | 82.31 | 29.43 |
| | Tibia | 60 | 20 | 18 | 31 | 6 | 33 | 25 | 16.5 | 100 | 157 | 43 | 88.95 | 25.18 |
| | Metatarsal | 76 | 25 | 19 | 19 | 0 | 33 | 25 | 16.5 | 100 | 150 | 29 | 80.66 | 29.17 |
| | Metapodial | 13 | 4 | 0 | 0 | 2 | 1 | 1 | 0.5 | 3 | 121 | 22 | 62.38 | 21.18 |
| | **Total** | **303** | | **89** | **104** | **12** | **132** | | **66** | | **157** | **22** | **79.48** | **27.01** |
| Riparo Tagliente Layer 35 SUNG | Humerus | 18 | 13 | 6 | 5 | 0 | 11 | 13 | 5.5 | 69 | 56 | 26 | 40.5 | 12.35 |
| | Radio-ulna | 5 | 4 | 4 | 1 | 0 | 9 | 10 | 4.5 | 56 | 74 | 18 | 51.63 | 18.42 |
| | Metacarpal | 32 | 23 | 4 | 3 | 2 | 16 | 19 | 8 | 100 | 88 | 14 | 39.31 | 13.58 |
| | Femur | 15 | 11 | 1 | 2 | 0 | 11 | 13 | 5.5 | 69 | 77 | 25 | 38.72 | 11.26 |
| | Tibia | 20 | 14 | 9 | 7 | 0 | 16 | 19 | 8 | 100 | 89 | 25 | 57.48 | 19.2 |
| | Metatarsal | 39 | 28 | 1 | 2 | 2 | 15 | 17 | 7.5 | 94 | 85 | 14 | 37.46 | 17.2 |
| | Metapodial | 9 | 7 | | | 0 | 8 | 9 | 4 | 50 | 73 | 15 | 40.67 | 12.55 |
| | **Total** | **138** | | **25** | **20** | **4** | **86** | | **43** | | **89** | **14** | **42.86** | **16.64** |
| Riparo Tagliente Layer 37 SUNG | Humerus | 18 | 10 | 9 | 4 | 0 | 9 | 17 | 4.5 | 82 | 65 | 19 | 59.6 | 9.65 |
| | Radio-ulna | 13 | 7 | 9 | 2 | 1 | 6 | 12 | 3 | 55 | 75 | 31 | 51.79 | 13.76 |
| | Metacarpal | 33 | 19 | 4 | 3 | 0 | 9 | 17 | 4.5 | 82 | 75 | 14 | 42.38 | 14.21 |
| | Femur | 22 | 13 | 4 | 10 | 0 | 5 | 10 | 2.5 | 45 | 55 | 22 | 42.86 | 13.75 |
| | Tibia | 39 | 22 | 15 | 16 | 2 | 11 | 21 | 5.5 | 100 | 99 | 29 | 50.23 | 16.24 |
| | Metatarsal | 30 | 17 | 2 | 2 | 1 | 10 | 19 | 5 | 91 | 95 | 14 | 41.38 | 15.99 |
| | Metapodial | 21 | 12 | | | 1 | 2 | 4 | 1 | 18 | 64 | 19 | 41.67 | 16.98 |
| | **Total** | **176** | | **43** | **37** | **5** | **52** | | **26** | | **99** | **14** | **44.45** | **15.21** |
| Abri du Maras Layer 4.1 MUNG | Humerus | 97 | 13 | 32 | 40 | 0 | 34 | 18 | 17 | 55 | 123 | 23 | 54.89 | 17.91 |
| | Radio-ulna | 123 | 17 | 34 | 32 | 1 | 41 | 21 | 20.5 | 66 | 193 | 22 | 59.64 | 26.36 |
| | Metacarpal | 58 | 8 | 13 | 14 | 2 | 19 | 10 | 9.5 | 31 | 181 | 16 | 60.25 | 34.07 |
| | Femur | 45 | 6 | 8 | 12 | 0 | 17 | 9 | 8.5 | 27 | 110 | 28 | 61.44 | 21.66 |
| | Tibia | 184 | 25 | 53 | 58 | 3 | 41 | 21 | 20.5 | 66 | 166 | 16 | 63.52 | 24.6 |
| | Metatarsal | 167 | 23 | 24 | 33 | 2 | 62 | 32 | 31 | 100 | 150 | 16 | 56.91 | 25.07 |
| | Metapodial | 68 | 9 | 1 | 1 | 3 | 11 | 6 | 5.5 | 18 | 101 | 16 | 45.19 | 20.92 |
| | **Total** | **742** | | **165** | **190** | **11** | **225** | | **112.5** | | **193** | **16** | **57.41** | **25.1** |
| Abri du Maras Layer 4.2 MUNG | Humerus | 41 | 11 | 13 | 21 | 0 | 20 | 12 | 10 | 44 | 130 | 32 | 62.66 | 21.92 |
| | Radio-ulna | 79 | 22 | 16 | 11 | 0 | 43 | 26 | 21.5 | 96 | 177 | 23 | 62.09 | 31.75 |
| | Metacarpal | 23 | 6 | 1 | 10 | 0 | 8 | 5 | 4 | 18 | 171 | 21 | 71.09 | 27.6 |
| | Femur | 32 | 9 | 10 | 10 | 0 | 15 | 9 | 7.5 | 33 | 105 | 26 | 72.41 | 32.54 |
| | Tibia | 95 | 26 | 32 | 40 | 2 | 45 | 27 | 22.5 | 100 | 161 | 34 | 76.97 | 36.1 |
| | Metatarsal | 72 | 20 | 10 | 7 | 0 | 29 | 17 | 14.5 | 64 | 219 | 23 | 59.77 | 23.89 |
| | Metapodial | 25 | 7 | 1 | 0 | 1 | 8 | 5 | 4 | 18 | 105 | 35 | 60.16 | 18.64 |
| | **Total** | **367** | | **83** | **99** | **3** | **168** | | **84** | | **219** | **21** | **66.9** | **29.88** |

*(Continued)*

**Table 2.** (Continued)

| Sites | Element | NISPa | % NISPa | Right | Left | juv | NME | % NME | MAU | % MAU | Max (mm) | Min (mm) | Mean (mm) | SD |
|---|---|---|---|---|---|---|---|---|---|---|---|---|---|---|
| Abri du Maras Layer 4.2 LUNG | Humerus | 32 | 16 | 11 | 11 | 0 | 16 | 16 | 8 | 35 | 147 | 48 | 92.06 | 23.92 |
| | Radio-ulna | 29 | 14 | 3 | 3 | 0 | 13 | 13 | 6.5 | 28 | 240 | 41 | 100.59 | 46.03 |
| | Metacarpal | 6 | 3 | 0 | 4 | 0 | 4 | 4 | 2 | 9 | 245 | 52 | 134 | 28.99 |
| | Femur | 29 | 14 | 7 | 6 | 0 | 14 | 14 | 7 | 30 | 149 | 27 | 95.93 | 33.85 |
| | Tibia | 93 | 46 | 25 | 35 | 1 | 46 | 47 | 23 | 100 | 309 | 28 | 101.67 | 49.13 |
| | Metatarsal | 8 | 4 | 1 | 1 | 0 | 3 | 3 | 1.5 | 7 | 132 | 39 | 77.88 | 34.25 |
| | Metapodial | 6 | 3 | 0 | 0 | 0 | 2 | 2 | 1 | 4 | 114 | 26 | 78 | 39.88 |
| | **Total** | **203** | | **47** | **60** | **1** | **98** | | **49** | | **309** | **26** | **98.52** | **42.49** |
| | Total | 2187 | | 531 | 599 | 43 | 881 | | 440.5 | | 309 | 9 | 67.53 | 9.14758412 |

metapodials undetermined and a horse metacarpal in layer 4.1 [29]. No complete bone circumference were preserved. For all the assemblages, most of the remains has a c1 circumference and a l1/l2 length (S2 Fig in S1 File).

Shaft fragments are very well represented for all the studied assemblages (Table 3). The epiphysis-diaphysis ratio is 0.17 (323/1917). The scarcity of epiphyses could be explained by several factors, such as differential conservation, anthropogenic breakage to recover bone grease and the use of bone cancellous as fuel [16, 127–129].

Regarding the dimensions of bone fragments, most of the remains are between 25 mm and 100 mm long (Tables 2 and 4, S3 Fig in S1 File). We note a differential fragmentation rate according to ungulate class size. None of the 4.2LUNG remains from Abri du Maras is smaller than 25 mm in length, whereas the SUNG remains from Riparo Tagliente are the most

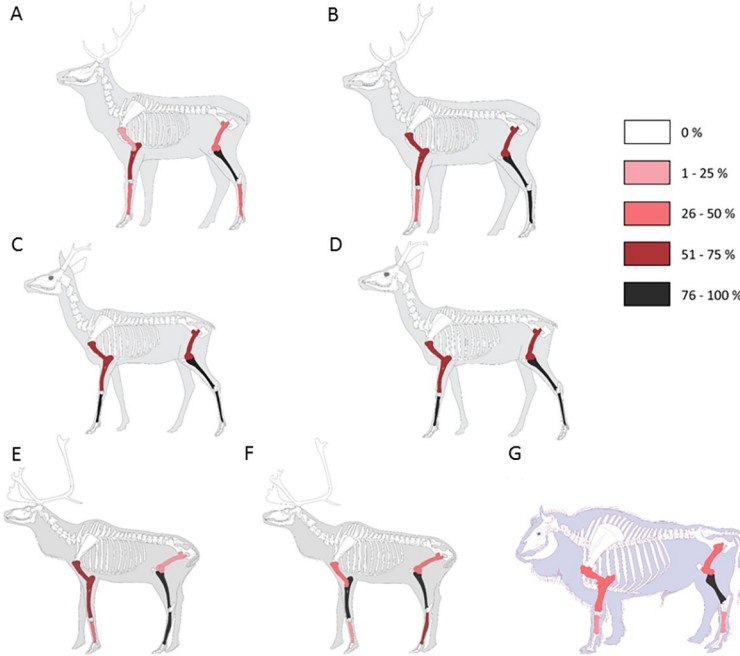

**Fig 2.** % MAU of the long bone elements for each assemblages: A- Saint Marcel g, B- Saint Marcel h, C- Riparo Tagliente 35, D- Riparo Tagliente 37, E- Abri du Maras 4.1, F & G- Abri du Maras 4.2.

**Table 3. Number of epiphyses and diaphyses long bone portions and ratios of epiphysis on diaphysis (E/D).**

| | Saint Marcel Cave | | Riparo Tagliente | | Abri du Maras | | |
| | g | h | 35 | 37 | 4.1 MUNG | 4.2 LUNG | 4.2 MUNG |
|---|---|---|---|---|---|---|---|
| Epiphysis | 25 | 41 | 13 | 24 | 162 | 27 | 31 |
| Diaphysis | 282 | 273 | 109 | 169 | 888 | 196 | 282 |
| **Epiphysis/Diaphysis** | **0.089** | **0.150** | **0.119** | **0.142** | **0.182** | **0.138** | **0.110** |

abundant for this length range. The SUNG remains from Riparo Tagliente 35 and 37 are never longer than 100 mm while almost 50% (89/203) of the remains from Abri du Maras 4.2LUNG are lengthier than 100 mm.

Fracture profiles are mainly curved/V-shaped with oblique angles and smooth edges. Green bone fractures are the most numerous, affecting more than 90% of the number of the studied remains for each layer. Dry bone fractures are recorded on approximatively 25% of all samples (Table 4). The number of indeterminate fractures varies from one site to another from 1 to 22%. Gelifraction is a minor factor in the fragmentation of bone assemblages, and only appears at Abri du Maras and Riparo Tagliente layer 35.

**Anthropogenic modifications.** Cutmarks are the best represented anthropogenic marks in our assemblages, followed by percussion marks (Table 4). We recorded a lower cutmarks rate in layers 4.1 and 4.2 of Abri du Maras than in the other sites. In most of the layers of Riparo Tagliente and Saint-Marcel Cave, we identified cutmarks on more than 50% of the identified fragments.

For the Abri du Maras and Riparo Tagliente samples, the number of burnt remains is very low (< 5%). However, in these two sites, we observe numerous indeterminate burnt splinters [106]. At Saint-Marcel, burnt elements are more abundant, with percentages between 15 to 35% among our samples (Table 4).

Apart from layer 4.1 of Abri du Maras, where no retouchers were identified regarding our sample, in the other studied samples their quantity varies according to the layers (Table 4). Saint-Marcel Cave is the site with the most bone retouchers (between 9 and 12%). At Riparo Tagliente, despite the predominance of small ungulates, percentages are around 6%.

**Percussion marks analysis.** *Percussion marks type.* Notches are the predominant percussion marks for the studied samples, followed by pits. Adhering flakes and grooves are the less numerous. Crushing marks are totally absent from our assemblages (Table 5).

**Table 4. Frost, green, dry and indeterminate long bone breakage (NISP and τ); measurement intervals along the length of the long bone (NR and τ); anthropogenic marks (NR and τ).**

| Sites | Saint Marcel Cave | | | | Riparo Tagliente | | | | Abri du Maras | | | | | |
| Layer | g | | h | | 35 | | 37 | | 4.1MUNG | | 4.2LUNG | | 4.2MUNG | |
| | NISP | % | NISP | % | NISP | % | NISP | % | NISP | % | NISP | % | NISP | % |
|---|---|---|---|---|---|---|---|---|---|---|---|---|---|---|
| **Green bone fracture** | 252 | 97.67 | 283 | 93.4 | 136 | 98.55 | 163 | 92.61 | 682 | 91.91 | 192 | 94.58 | 358 | 97.55 |
| **Dry bone fracture** | 71 | 27.52 | 152 | 50.17 | 34 | 24.64 | 20 | 11.36 | 222 | 29.92 | 53 | 26.11 | 75 | 20.44 |
| **Und** | 2 | 0.78 | 4 | 1.32 | 9 | 6.52 | 16 | 9.09 | 88 | 11.86 | 45 | 22.17 | 61 | 16.62 |
| **Gelifraction** | – | – | - | - | 5 | 3.62 | – | – | 17 | 2.29 | 8 | 3.94 | 38 | 10.35 |
| **Nr burnt** | 40 | 15.50 | 107 | 35.31 | 7 | 5.07 | 4 | 2.27 | 10 | 1.35 | 1 | 0.49 | 1 | 0.27 |
| **Cutmarks** | 175 | 68.36 | 166 | 54.97 | 64 | 48.12 | 91 | 52.9 | 211 | 37.54 | 59 | 50.86 | 83 | 34.02 |
| **Percussion marks** | 146 | 56.59 | 136 | 44.88 | 56 | 40.58 | 69 | 39.2 | 197 | 26.55 | 77 | 37.93 | 112 | 30.52 |
| **Retoucher** | 23 | 8.98 | 37 | 12.25 | 8 | 6.0 | 11 | 6.40 | - | - | 5 | 4 | 6 | 2.46 |
| **Total NISP** | **258** | **100** | **303** | **100** | **138** | **100%** | **176** | **100** | **742** | **100** | **203** | **100** | **367** | **100** |

**Table 5. Number of percussion marks by layer of the studied sites, by type of percussion mark and by element (each percentage has been calculated on the total NR).**

| Site | Saint Marcel Cave | | | | Riparo Tagliente | | | | Abri du Maras | | | | | |
|---|---|---|---|---|---|---|---|---|---|---|---|---|---|---|
| Layer | g | | h | | 35 | | 37 | | 4.1MUNG | | 4.2LUNG | | 4.2MUNG | |
| | NISP | % | NISP | % | NISP | % | NISP | % | NISP | % | NISP | % | NISP | % |
| **Percussion marks** | 253 | | 242 | | 88 | | 107 | | 219 | | 97 | | 148 | |
| **Notch** | 171 | 67.6% | 151 | 62.4% | 58 | 65.9% | 90 | 84.1% | 218 | 99.5% | 93 | 95.9% | 144 | 97.3% |
| triangular | 21 | 12.3% | 25 | 16.6% | 6 | 10.3% | 0 | 0% | 0 | 0% | 0 | 0% | 0 | 0% |
| ovoid | 150 | 87.7% | 126 | 83.4% | 52 | 89.7% | 90 | 100% | 0 | 0% | 94 | 101% | 144 | 100% |
| **Pit** | 59 | 23.3% | 69 | 28.5% | 29 | 33.0% | 16 | 15.0% | 1 | 0.5% | 4 | 4.1% | 2 | 1.4% |
| triangular | 32 | 54.2% | 57 | 82.6% | 9 | 31.0% | 7 | 43.8% | 0 | 0% | 0 | 0% | 0 | 0% |
| ovoid | 27 | 45.8% | 12 | 17.4% | 20 | 69.0% | 9 | 56.3% | 1 | 100% | 4 | 100% | 2 | 100% |
| **Groove** | 2 | 0.8% | 5 | 2.1% | 1 | 1.1% | 0 | 0% | 0 | 0% | 0 | 0% | 0 | 0% |
| **Crushing marks** | 0 | 0% | 3 | 1.2% | 0 | 0% | 0 | 0% | 0 | 0% | 0 | 0% | 0 | 0% |
| **Adhering flake** | 21 | 8.3% | 17 | 7% | 0 | 0% | 1 | 0.9% | 0 | 0% | 10 | 10.3% | 2 | 1.4% |
| Humerus | 23 | 9.1% | 22 | 9.1% | 10 | 11.4% | 25 | 23.4% | 44 | 20.1% | 22 | 22.7% | 19 | 12.8% |
| Radio-ulna | 40 | 15.8% | 23 | 9.5% | 3 | 3.4% | 3 | 2.8% | 24 | 11.0% | 11 | 11.3% | 26 | 17.6% |
| Metacarpal | 61 | 24.1% | 47 | 19.4% | 19 | 21.6% | 15 | 14% | 14 | 6.4% | 3 | 3.1% | 8 | 5.4% |
| Femur | 26 | 10.3% | 26 | 10.7% | 13 | 14.8% | 14 | 13.1% | 21 | 9.6% | 26 | 26.8% | 15 | 10.1% |
| Tibia | 55 | 21.7% | 38 | 15.7% | 12 | 13.6% | 25 | 23.4% | 77 | 35.2% | 34 | 35.1% | 56 | 37.8% |
| Metatarsal | 31 | 12.3% | 78 | 32.2% | 29 | 33% | 22 | 20.6% | 37 | 16.9% | 0 | 0% | 20 | 13.5% |
| Metapodial | 17 | 6.7% | 8 | 3.3% | 2 | 2.3% | 3 | 2.8% | 2 | 0.9% | 1 | 1.0% | 4 | 2.7% |
| **Total** | 253 | | 242 | | 88 | | 107 | | 219 | | 97 | | 148 | |

In the Abri du Maras assemblages, almost all the percussion marks are notches, but we also observed pits with ovoid morphology and adhering flakes. A slightly higher number of pits and adhering flakes was identified for 4.2LUNG layer than for layer 4.1. No grooves were observed (Table 5).

In the Saint-Marcel assemblages (Table 5), around two-thirds of the percussion marks are notches and around a quarter are pits. The rate of adhering flakes and grooves is lower, at around 10%. Most of the notches, in both layers, are conchoidal and a minority present an internal triangular pit or groove. Triangular percussion pits are more numerous in both layers.

In the Riparo Tagliente layers, grooves and adhering flakes constitute around 1% of all percussion marks. Again, notches are the most numerous types of marks identified. They are all conchoidal whereas a minority identified in layer 35 present an internal triangular pit or a groove. Pits are the second most represented type of mark, generally with an ovoid morphology (Table 5).

**The number of percussion marks by éléments.** At Abri du Maras, the three faunal samples show a relatively similar distribution of percussion marks per element (Table 5). One third of the percussion marks are located on tibias, while only a few percussion marks were identified on metacarpal remains. More specifically, for layer 4.2LUNG, none of the metatarsal remains present percussion marks. Percussion marks are equally represented for the other long bones.

Similarly, at Saint-Marcel Cave, percussion mark distribution is equivalent per element for both levels. Tibias and metapodials are the most impacted fragments.

For the Riparo Tagliente layers, tibias and metapodials are also the bone elements with the most marked areas.

The number of percussion marks and NISPa by element are significantly positively correlated in Abri du Maras 4.2LUNG (Spearman correlation: p-value = 0.032, $r_s$ = 0.82), Abri du

Maras 4.2MUNG (Spearman correlation: df = 6, p-value < 0.01, $r_s$ = 0.96) and Saint Marcel Cave level h (Spearman correlation: df = 6, p-value = 0.007, rs = 0.93) and Riparo Tagliente 35 (Spearman correlation: df = 6, p-value = 0.012, $r_s$ = 0.86). In the other assemblages, they were not significantly correlated (Abri du Maras, 4.1; Saint Marcel Cave, g and Riparo Tagliente, 37: Spearman correlation: df = 6, p-value > 0.05). In terms of the whole assemblage of each site, only Saint Marcel Cave showed a positively significant correlation between these variables, while the Abri du Maras and Riparo Tagliente faunal assemblages did not show a correlation (Spearman correlation: df = 6, p-values > 0.05).

**The percussion marks frequency by bone areas.** The frequencies of the impacted portions and long bone density indices are not significantly correlated for each assemblage (Spearman correlation: df = 25; p-value > 0.05). None of the studied assemblages or long bones show a systematic distribution of percussion marks, i.e., only one or two struck areas (Figs 3–8). The number of areas with at least one percussion mark varied between two and 14 areas per bone element. In order to identify tendencies, we took into account frequencies equal to or higher than 0.33 and counted their occurrence per element, so that we could carry out inter- and intra-site comparisons. The lateral side of the medial portion (p3l) of the humerus shows higher proportions in five faunal assemblages (layers g, 35, 37, 4.1 and 4.2MUNG, 71.4%) (Fig 3). We record high frequencies for the femur in five assemblages on the proximal portion (p2l) of the lateral side (layers g, h, 35, 4.1 and 4.2LUNG; 71.4%) (Fig 6). For the radio-ulna, the most impacted area is the anterior side of the distal portion (p4a) (layers g, h, 4.1 and 4.2LUNG; 42.9%) (Fig 4). For the metacarpal, we identified two areas with high frequencies of impact traces: the lateral side of the proximal portion (p2l) and the medial side of the medial portion (p3m), for three of our samples (layers g, h, and 4.1 and layers g, 4.1 and 4.2MUNG; 42.9%) (Fig 5). On the metatarsal, five areas were identified. Two tibia series do not show high frequencies (Saint-Marcel Cave layer g and Abri du Maras layer 4.2LUNG) (Fig 8). For the others, only 28.5% show a tendency with higher frequencies. The areas with higher frequency tendencies are always located on the diaphysis. Within the same site, the most impacted areas does not systematically display similar locations.

**Non-random and counter-intuitive distribution of percussion marks.** We compare our results with those from [43] two Middle Palaeolithic Spanish sites (MIS 5–9): Bolomor (levels IV, IX, XIIa and XIIc) and Gran Dolina (TD10-1). For the experimental data, we used the intuitive pattern presented in [122] (Figs 9–11).

The hypothesis tested on the archaeological samples, including comparative data, is the presence of a random pattern of percussion mark distribution. When data are quantitatively limited, we group the assemblages from the same site together. The results show that at Bolomor, percussion mark areas are not randomly distributed, in contrast to Gran Dolina with the previously published chi-square. Regarding our studied sites, Riparo Tagliente presents a probable random distribution of the marked areas (Chi-square: p-value > 0.05). Nonetheless, in the site of Abri du Maras, four long bones: humerus (Chi-square: p-value < 0.001, $\chi^2$ = 40.86), radio-ulna (Chi-square: p-value = 0.002, $\chi^2$ = 29.53), femur (Chi-square: p-value < 0.001, $\chi^2$ = 21.11) and metatarsal (Chi-square: p-value = 0.001, $\chi^2$ = 31.08) show that bone areas are not equally impacted. At Saint-Marcel Cave, three long bones: tibia (Chi-square: p-value = 0.031, $\chi^2$ = 21.16), metacarpal (Chi-square: p-value = 0.019, $\chi^2$ = 22.76) and metatarsal (Chi-square: p-value < 0.001, $\chi^2$ = 40.06) yield similar test results.

In order to develop the analysis at an intra-site scale, we need to test each assemblage independently. For the sake of representativeness, we only considered assemblages with at least 15 areas with percussion marks (Figs 9–11). Except the radio-ulna and the metatarsal of Abri du Maras 4.2LUNG, all the elements from the Abri du Maras and Saint-Marcel Cave levels were eligible to be tested. The three Abri du Maras assemblages show differences. The level 4.1

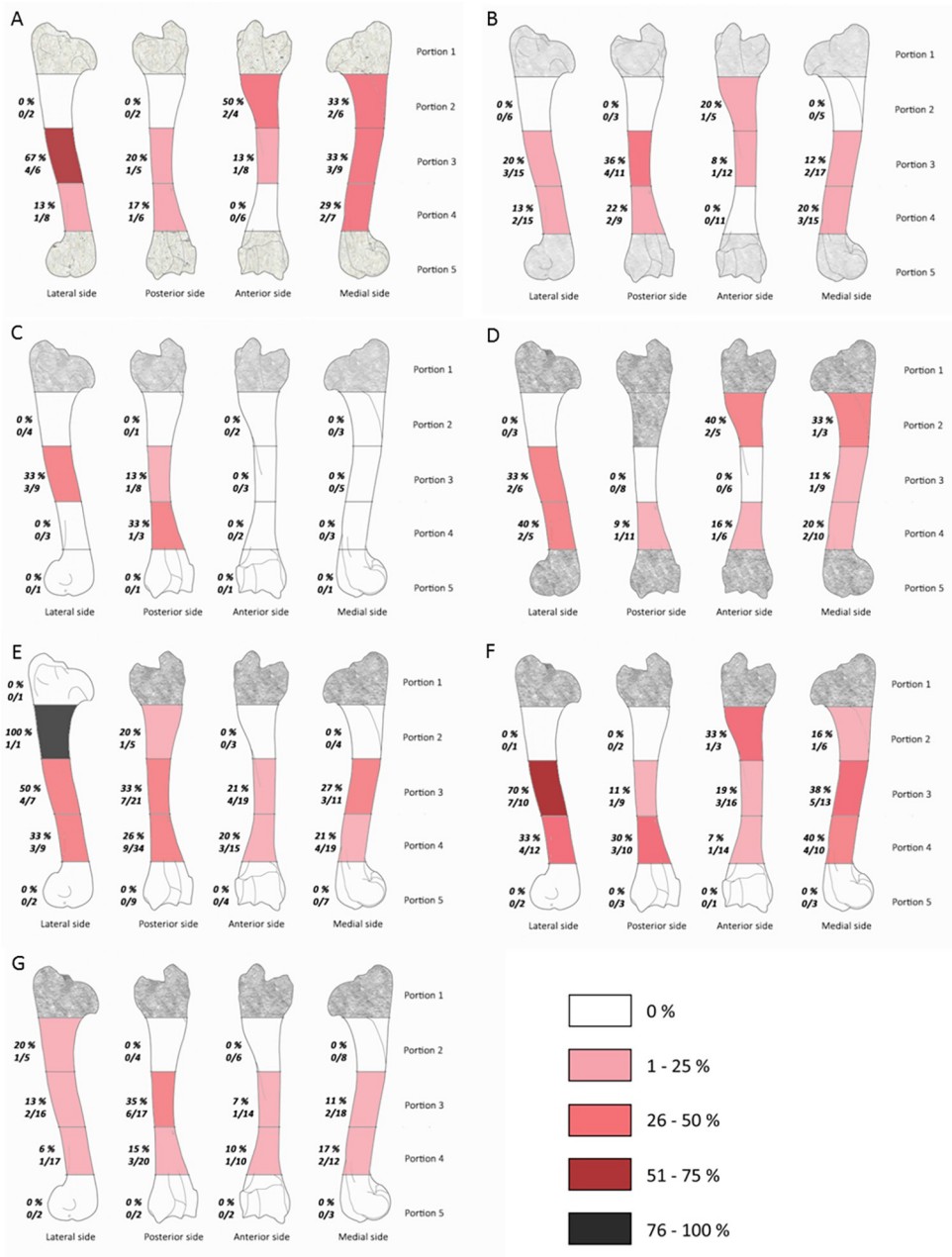

**Fig 3.** Frequencies of percussion marks (% and Number of area with percussion mark(s) / Number of preserved area) by portion on humerus from Saint Marcel Cave level g (A) and level h (B), Riparo Tagliente level 35 (C) and level 37 (D), and Abri du Maras level 4.1 (E), level 4.2 LUNG (F), level 4.2 MUNG. The absence of portion in each assemblage is grey.

sample presents a non-random distribution of the impacted areas for the humerus (Chi-square: p-value < 0.001, $\chi^2$ = 29.11), the femur (Chi-square: p-value = 0.002, $\chi^2$ = 32) and the metatarsal (Chi-square: p-value < 0.001, $\chi^2$ = 38). In level 4.2, the radio-ulnas of the 4.2MUNG assemblage (Chi-square: p-value = 0.001, $\chi^2$ = 30.74) and the humeri of the 4.2LUNG sample (Chi-square: p-value = 0.003, $\chi^2$ = 21.2) present proportions of bone areas unequally marked by percussion. The Saint-Marcel Cave assemblages show differences

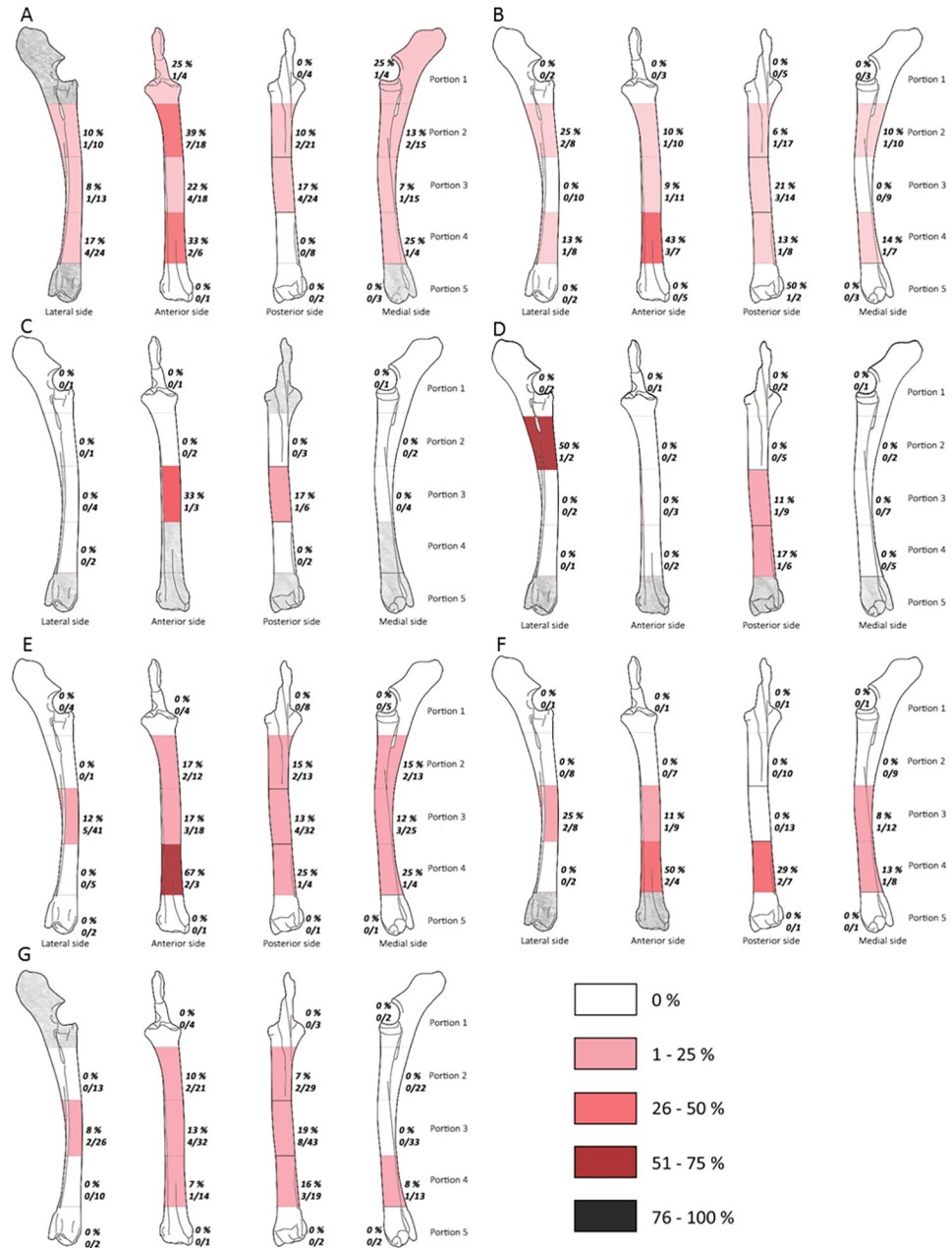

**Fig 4.** Frequencies of percussion marks (% and Number of area with percussion mark(s) / Number of preserved area) by portion on radio-ulna from Saint Marcel Cave level g (A) and level h (B), Riparo Tagliente level 35 (C) and level 37 (D), and Abri du Maras level 4.1 (E), level 4.2 LUNG (F), level 4.2 MUNG. The absence of portion in each assemblage is grey.

depending on the element. The metatarsal for both assemblages present a non-random distribution (Chi-square: level g: p-value = 0.02, $\chi^2$ = 22 and level h: p-value < 0.001, $\chi^2$ = 35.73). In layer g, percussion marks are randomly distributed on the metacarpal (Chi-square: p-value = 0.01, $\chi^2$ = 24.64), whereas this is not the case in level h. Finally, bone areas are equally marked for the tibias from each level.

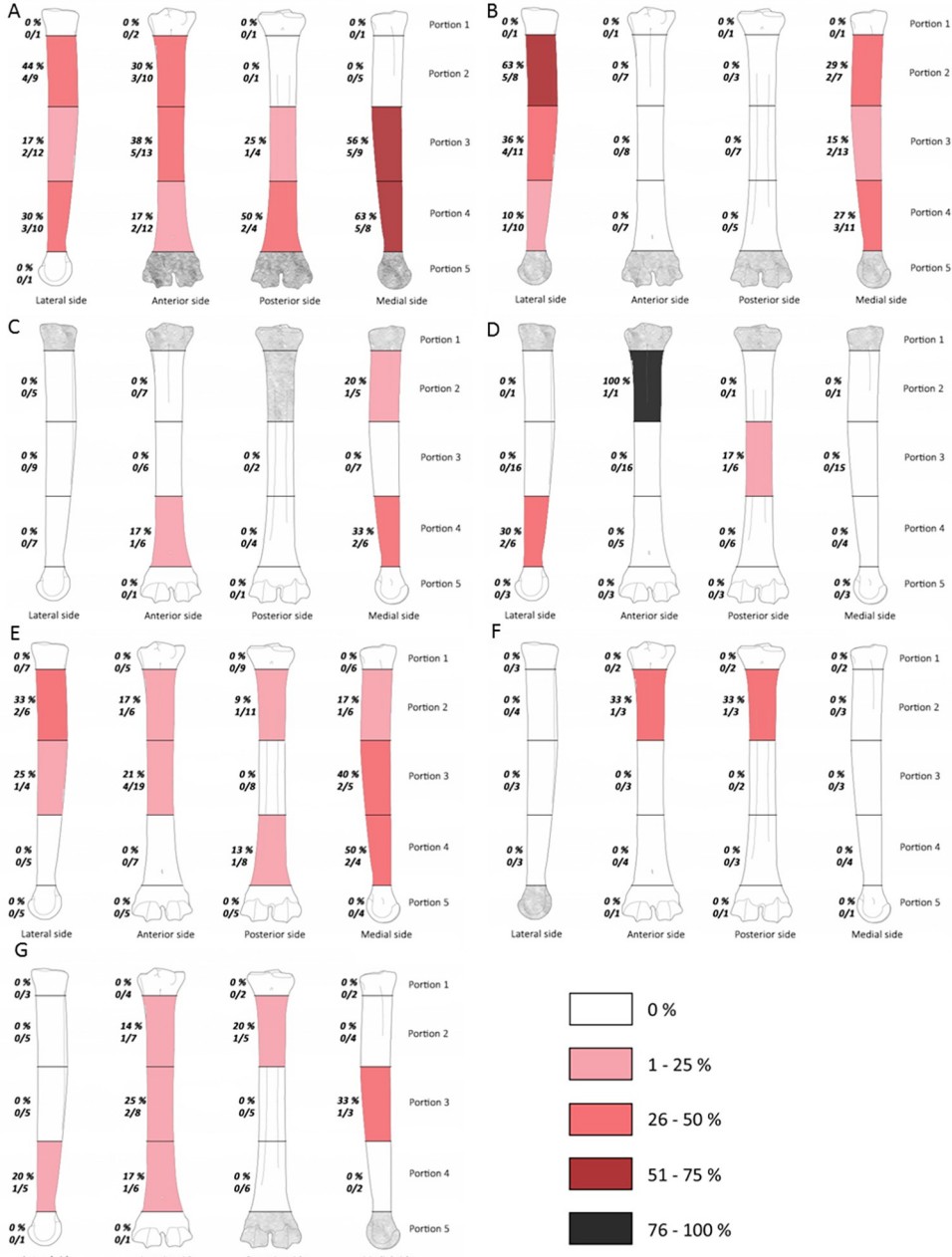

**Fig 5.** Frequencies of percussion marks (% and Number of area with percussion mark(s) / Number of preserved area) by portion on metacarpal from Saint Marcel Cave level g (A) and level h (B), Riparo Tagliente level 35 (C) and level 37 (D), and Abri du Maras level 4.1 (E), level 4.2 LUNG (F), level 4.2 MUNG. The absence of portion in each assemblage is grey.

If we only consider the assemblages with non-random distribution, it is possible to discuss butchery traditions or intuitive patterns. Some works highlight the existence of intuitive patterns in percussion mark distribution for marrow recovery, based on experiments involving non-trained experimenters [45, 46, 122]. Using the results of this experiment and chi-square analysis, we compared whether the non-random distribution of percussion marks is statistically different or not from intuitive pattern distribution.

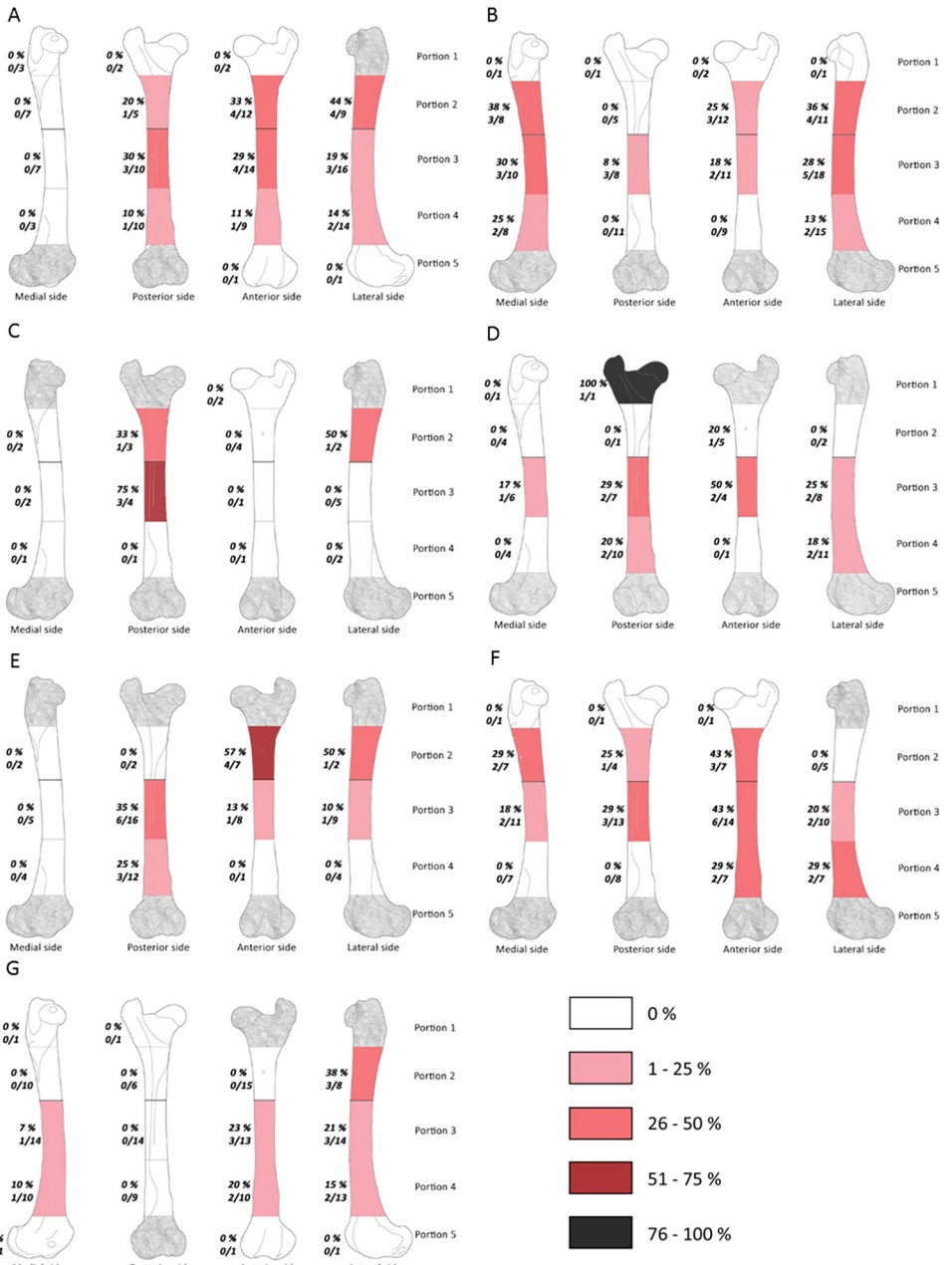

**Fig 6.** Frequencies of percussion marks (% and Number of area with percussion mark(s) / Number of preserved area) by portion on femur from Saint Marcel Cave level g (A) and level h (B), Riparo Tagliente level 35 (C) and level 37 (D), and Abri du Maras level 4.1 (E), level 4.2 LUNG (F), level 4.2 MUNG. The absence of portion in each assemblage is grey.

The Bolomor site reveals a different distribution from the intuitive one (Chi-square: p-value < 0.01, humerus: $\chi^2 = 158.58$; femur: $\chi^2 = 39.52$; tibia: $\chi^2 = 39.57$), except for the radio-ulna (Chi-square: p-value > 0.05, $\chi^2 = 19.3$). For radio-ulnas, the same results were noted as for radio-ulnas [122]. In the Abri du Maras and Saint-Marcel Cave assemblages, only the elements with non-random percussion mark distribution were tested with intuitive patterns. All of them show a different distribution to the intuitive pattern (Chi-square: Maras: humerus, p-

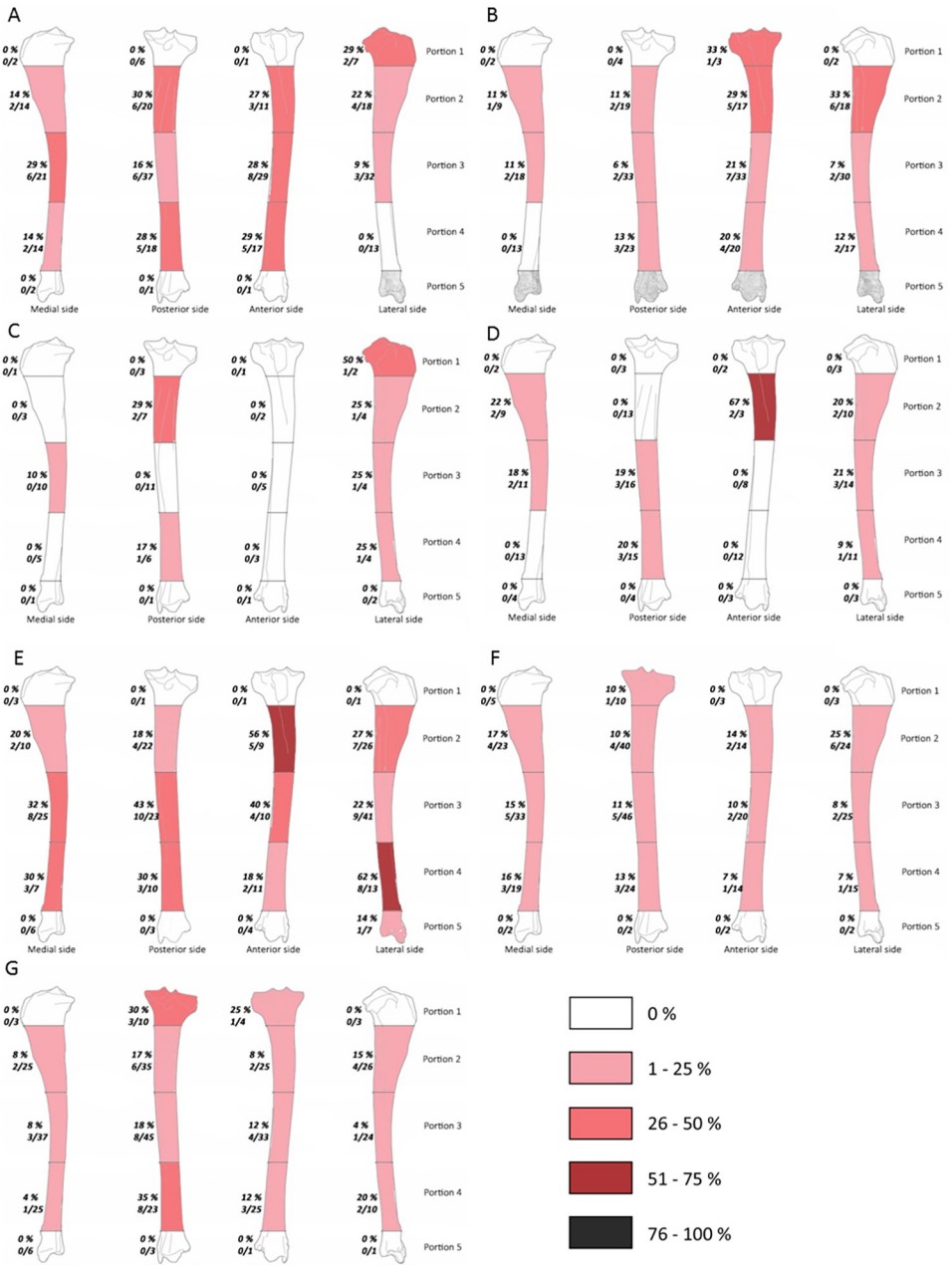

**Fig 7.** Frequencies of percussion marks (% and Number of area with percussion mark(s) / Number of preserved area) by portion on humerus from Saint Marcel Cave level g (A) and level h (B), Riparo Tagliente level 35 (C) and level 37 (D), and Abri du Maras level 4.1 (E), level 4.2 LUNG (F), level 4.2 MUNG. The absence of portion in each assemblage is grey.

value < 0.001, $\chi^2$ = 51.86; radius, p-value = 0.033, $\chi^2$ = 21.01 and femur, p-value = 0.006, $\chi^2$ = 26.1; Saint-Marcel: tibia, p-value = 0.008, $\chi^2$ = 25.53). at Abri du Maras, the elements with results showing unequal marked areas also display a different distribution from an intuitive pattern (Chi-square: Abri du Maras, 4.1: humerus, p-value < 0.001, $\chi^2$ = 38.43 and femur, p-value < 0.001, $\chi^2$ = 38.82; 4.2MUNG: radius, p-value < 0.001, $\chi^2$ = 22.69 and 4.2LUNG: humerus, p-value = 0.002, $\chi^2$ = 21.96).

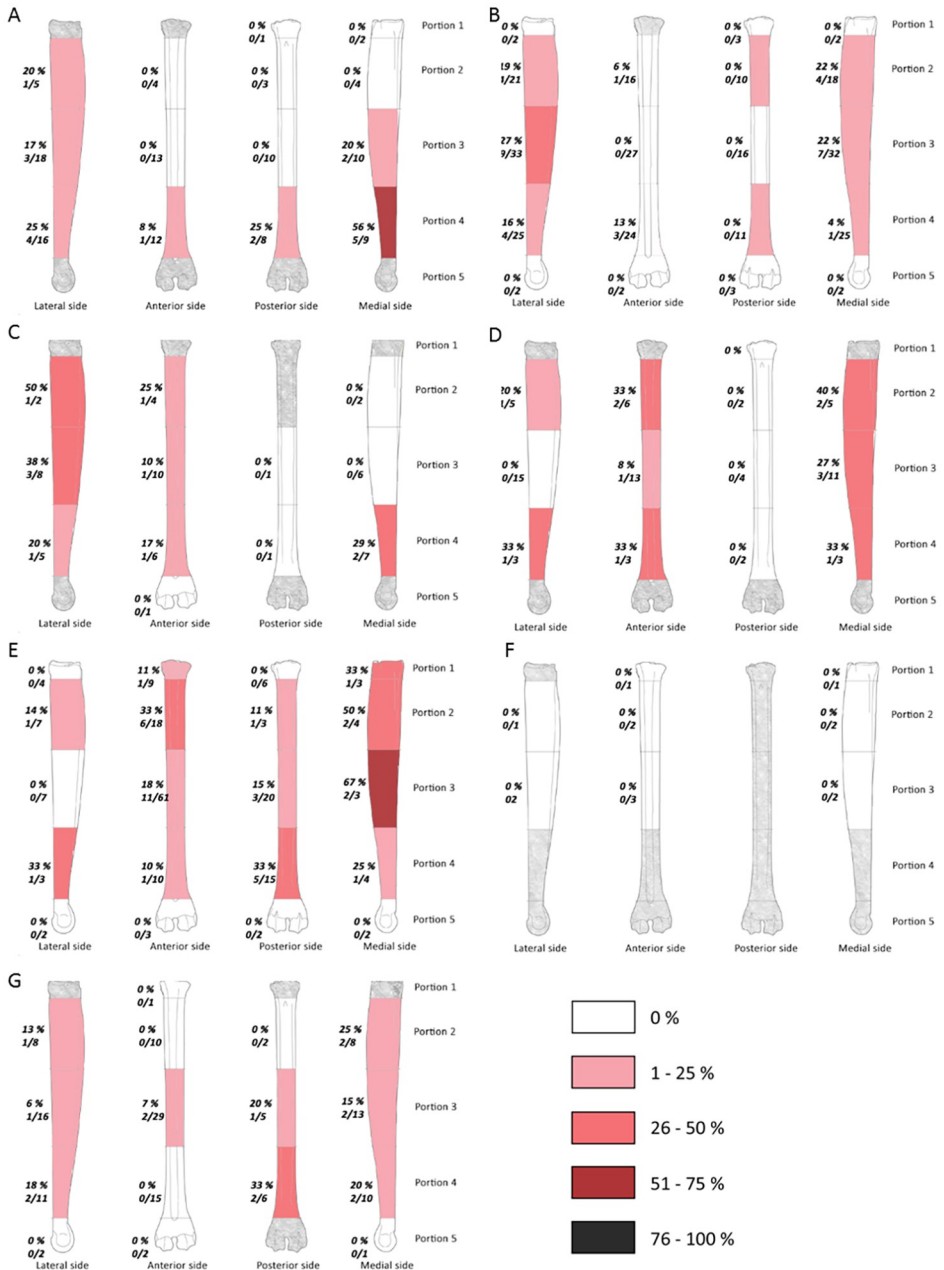

**Fig 8.** Frequencies of percussion marks by portion on metatarsal of each sample (% and Number of area with percussion mark(s) / Number of preserved area); Saint Marcel Cave level g (A), level h (B); Riparo Tagliente level 35 (C), level 37 (D); Abri du Maras level 4.1 (E), level 4.2 LUNG (F), level 4.2 MUNG. The absence of portion in each assemblage is grey.

**Significant differences between assemblages.** The MCA includes both the bone areas with and without percussion marks (S10 Fig in S1 File). The first axis, accounting for around 17% of the total variance, is related to the impact traces presence, with impacted areas on the right of the chart and non-impacted areas on the left. Predictably, the experiment yielded more preserved and impacted areas than the archaeological assemblages did. Also, the most impacted portion and side seem to be respectively portion 3 and the medial side. In the

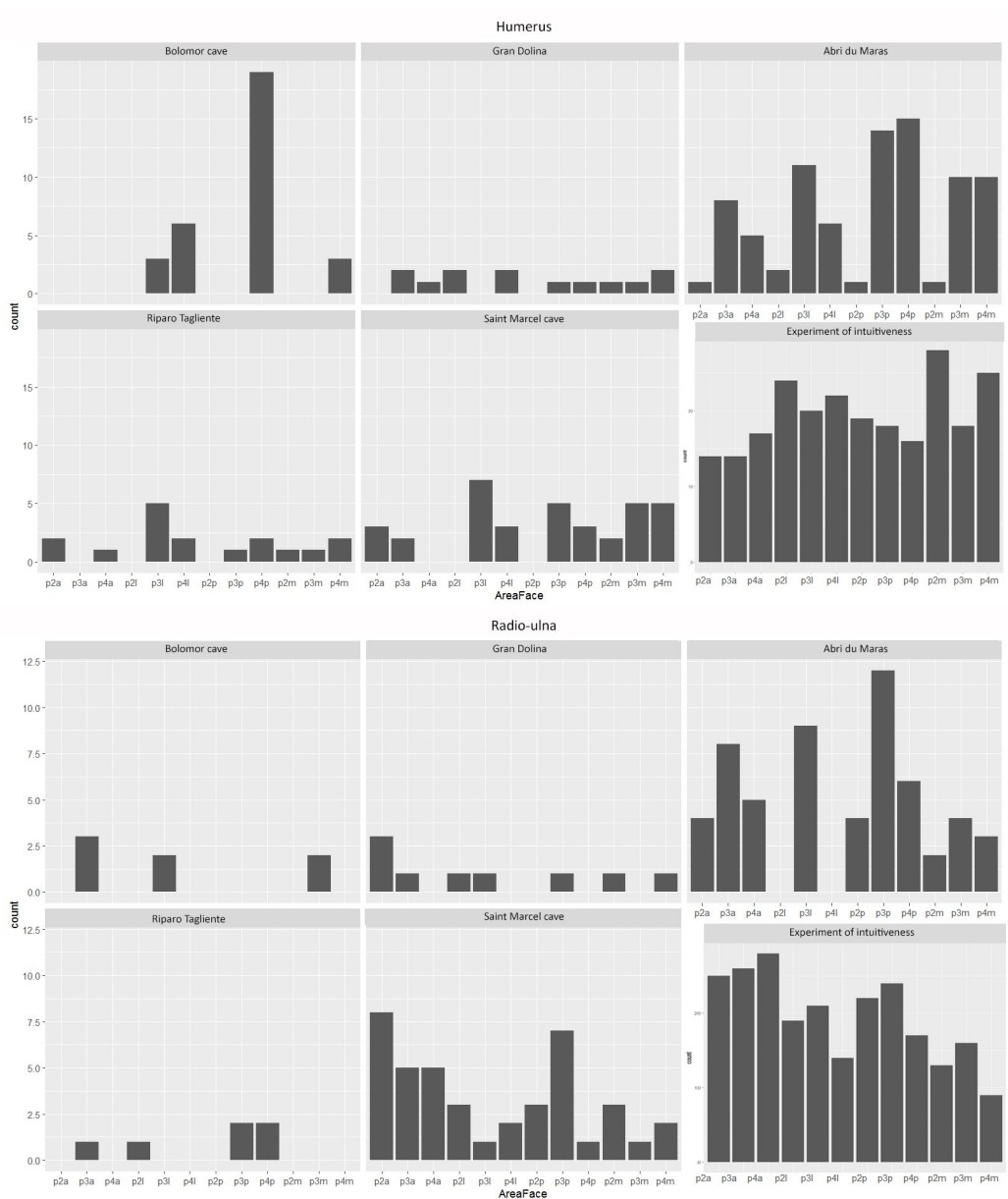

**Fig 9. Number of areas with percussion marks for each archaeological site and for the intuitive experiment model for humerus and radio-ulnas.** Portion 2 (P2): proximal diaphysis; Portion 3 (P3): medial diaphysis; Portion 4 (P4): distal diaphysis; sides: anterior (a), lateral (l), posterior (p) and medial (m).

controlled experiment, all the remains were kept and conserved, whereas several remains are missing from the archaeological assemblages due to differential conservation or problems of identification (S2 Table in S1 File) (Figs 9–11).

In order to examine in more detail, the relation between the portions and the sides bearing percussion marks depending on the sites and bone elements, we excluded areas without percussion marks from the analysis. The location of percussion marks on the portions and sides are used to build the axes of the MCA while the sites and their levels are displayed as illustrative variables. Separate MCA are computed for each bone element (S10 and S11 Figs in S1 File). In most of the plots, most of the sites are very close to the centre of the chart, showing that it is

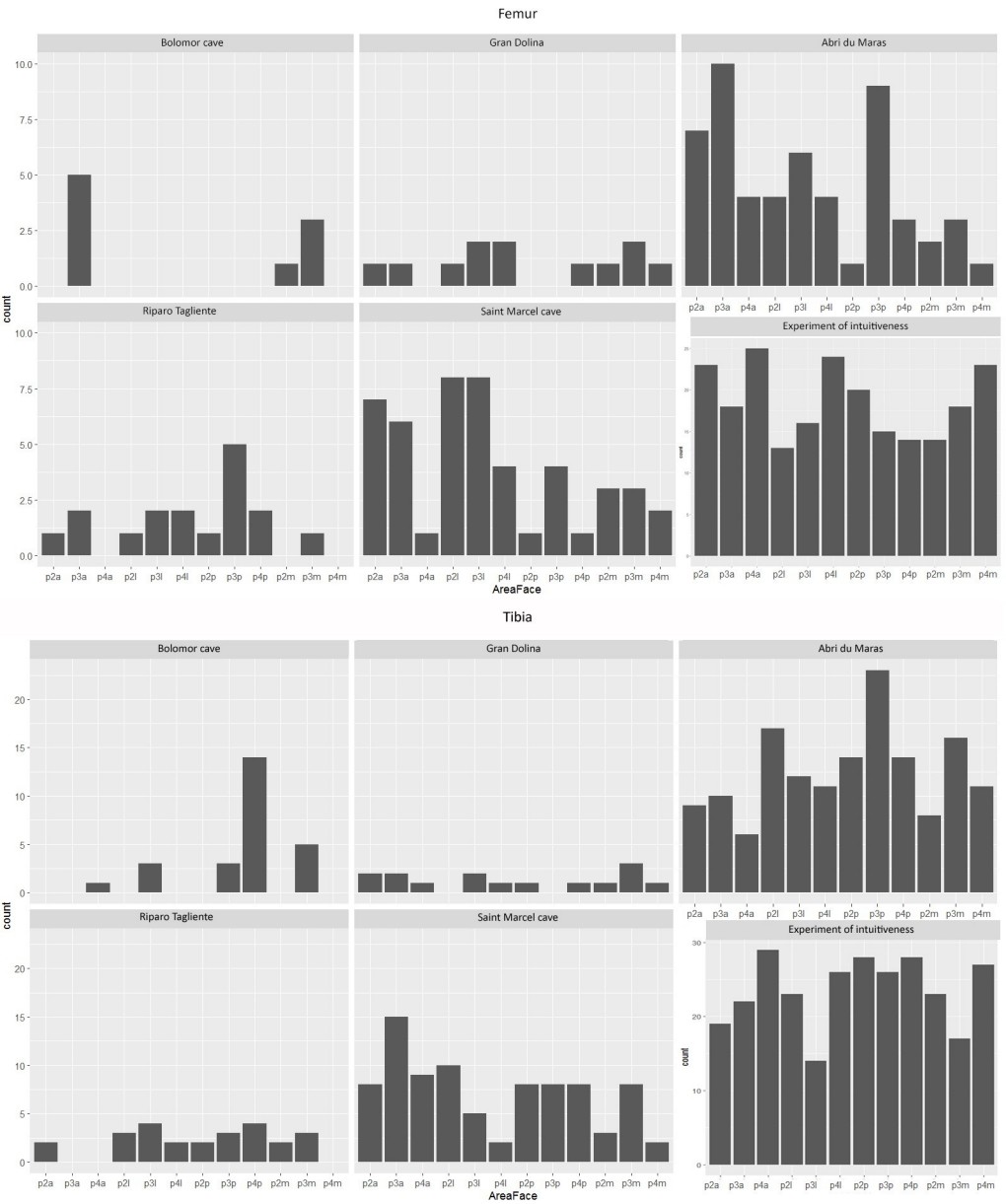

**Fig 10. Number of areas with percussion marks for each archaeological site and for the intuitive experiment model for femora and tibias.** Portion 2 (P2): proximal diaphysis; Portion 3 (P3): medial diaphysis; Portion 4 (P4): distal diaphysis; sides: anterior (a), lateral (l), posterior (p) and medial (m).

difficult to differentiate the sites and the experiments based on these variables. For the humerus plot, the Abri du Maras seems to be different to the intuitive experiment, but the series are close to the other sites. On the radio-ulna chart, the Abri du Maras and Saint-Marcel series show some differences, but they are encompassed in the ellipse of Gran Dolina and Riparo Tagliente. Regarding the femur, the ellipses of the Saint-Marcel, Abri du Maras and Riparo Tagliente assemblages intersect but they present differences. Nevertheless, the series of Saint-Marcel and Abri du Maras are included in the ellipse of Gran Dolina. However, the site of Bolomor is always far removed from the other experimental and archaeological groups.

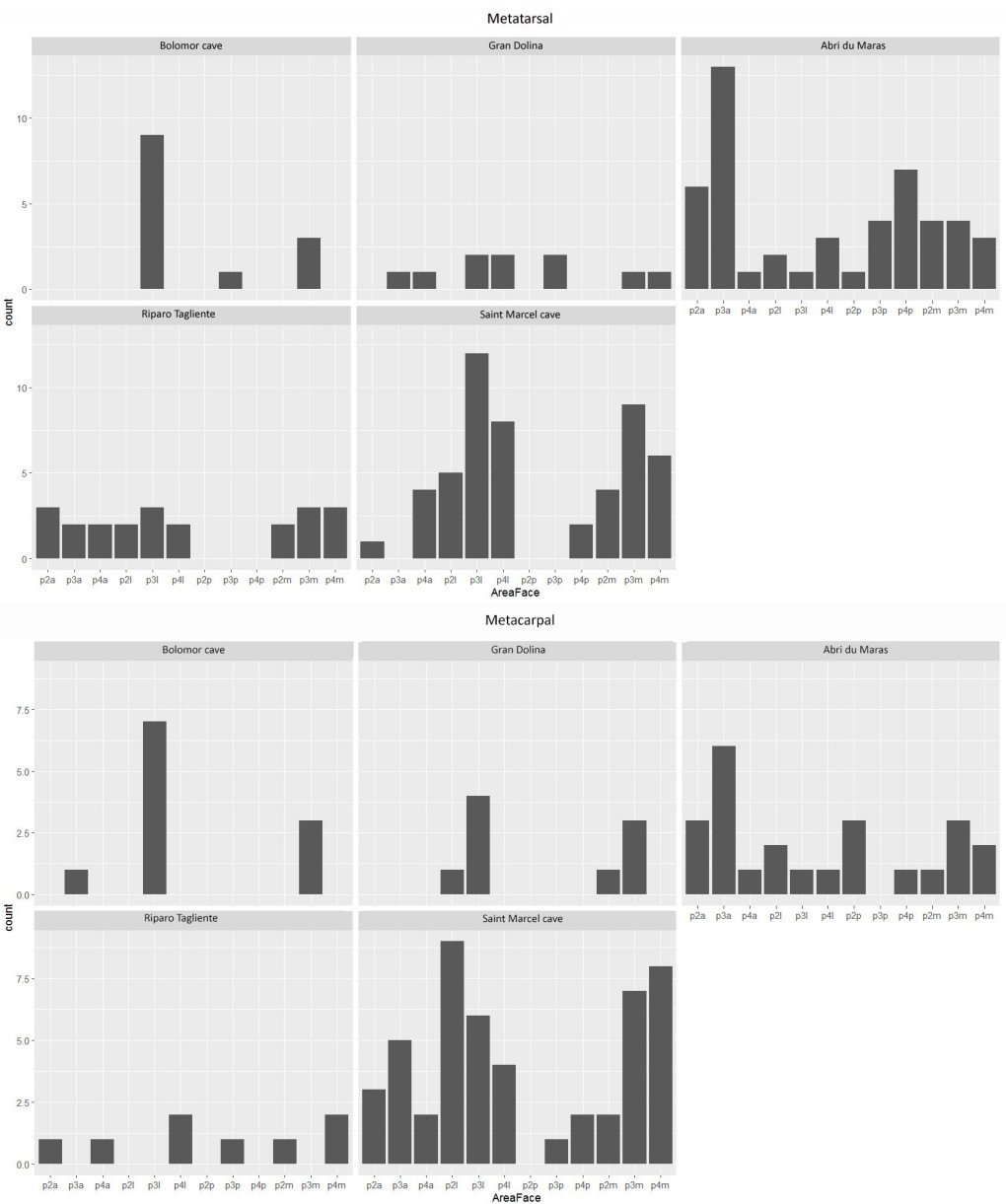

**Fig 11. Number of areas with percussion marks for each archaeological site for metacarpals and metatarsals.** Portion 2 (P2): proximal diaphysis; Portion 3 (P3): Medial diaphysis; Portion 4 (P4): Distal diaphysis; sides: anterior (a), lateral (l), posterior (p) and medial (m).

Regarding the MCA including experimental data, we observe that the batting experiment is distant from the other experiments [43, 130]. The most impacted areas are on radio-ulna, the portion 2 and posterior side, on femur the portion 2 and anterior side and on tibia with the more impacted area the portion 2 and lateral side (lateral side has a low $\cos^2$) (Figs 10 and 11). The other experiment of [43] is quite distant from the center (humerus, radio-ulna and femur), but less than the batting series. The other experimental assemblages are grouped towards the plot center, showing that the considered variables (the portion or the side) are not discriminatory. Besides, the sample from Riparo Tagliente level 35 is distant from the other archaeological assemblages for the femur and humerus (Fig 12).

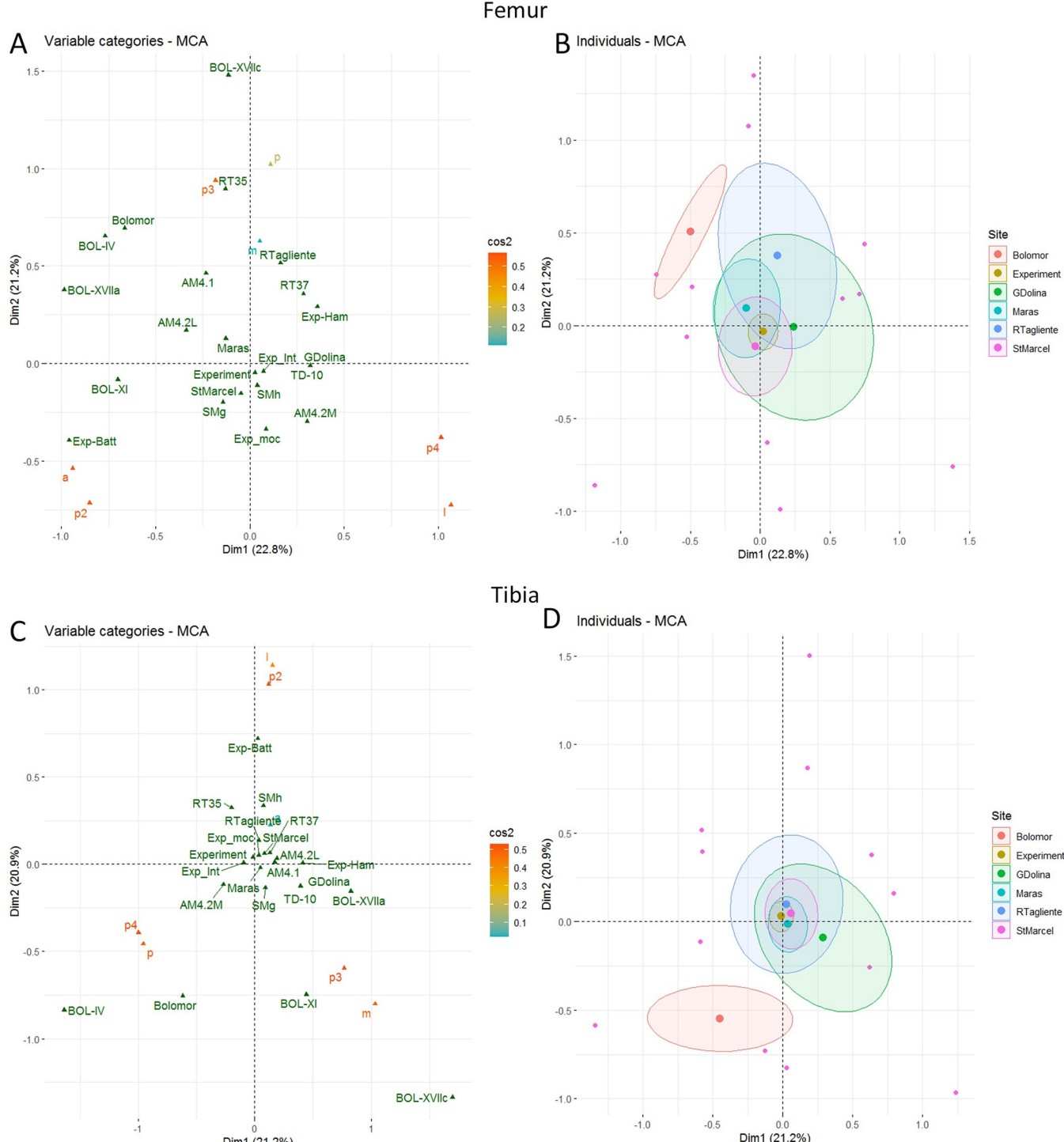

**Fig 12.** MCA of the areas with percussion marks by bone portion and side for femur (A-B) and tibia (C-D). Sites are included as illustrative variable only. Cos$^2$ are displayed following a colour gradient.

Regarding the metapodials, the absence of experiments seems to influence plot distribution (Fig 13). For the metapodial charts, Bolomor shows a certain proximity to the Gran Dolina assemblage with more impact traces on portion 3 and the anterior side (but the anterior side

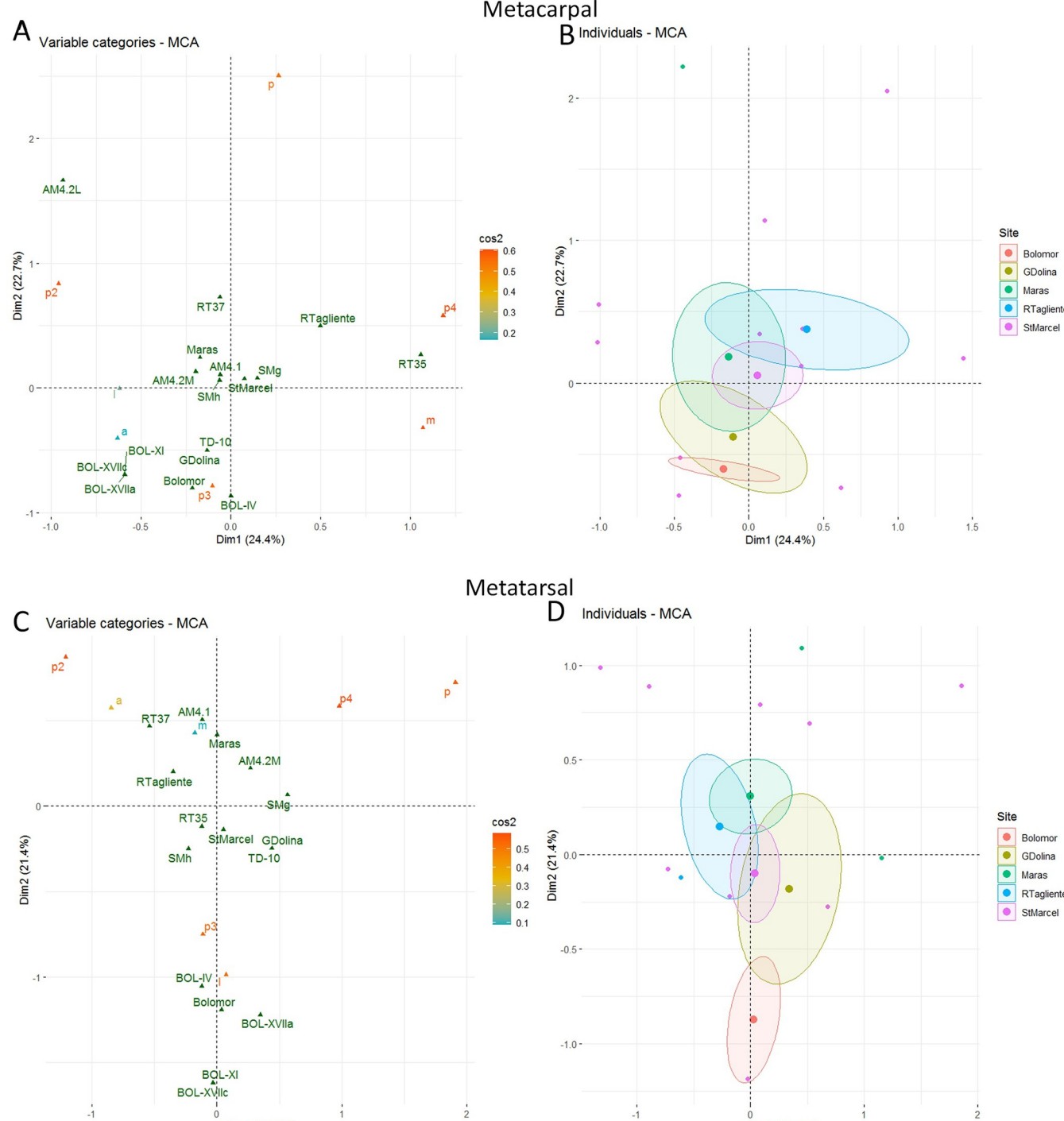

**Fig 13.** MCA of the areas with percussion marks by bone portion and side for metacarpal (A-B) and metatarsal (C-D). Sites are included as illustrative variable only. Cos$^2$ are displayed following a colour gradient.

has a low $\cos^2$) on the metacarpal and on portion 3 and the lateral side on the metatarsal. Bolomor is set apart from our series.

In order to test if the tendencies observed in the MCA are statistically significant (in other words, if they could be due to chance or not), we used the Fisher exact test. The results display significant differences for the area with impact distribution on the metatarsal between Abri du Maras and Saint-Marcel (p3a/p3l, p-value < 0.001 and p3a/p4l, p-value < 0.001).

## Discussion

Through the analysis of seven faunal assemblages from three Southern Europe Middle Palaeolithic sites: Abri du Maras, Saint Marcel and Riparo Tagliente, we conducted a large comparative analysis of the percussion marks distribution on long bones. In all these assemblages, we observed distinct and homogeneous long bone fragmentation. We also identified a high number of green bone fractures and several percussion marks of different types (notches, adhering flakes, pits and grooves) as the result of anthropogenic breakage. The sites record intensive human occupation, and the number and diversity of the percussion marks attest to systematic long bone breakage for marrow extraction.

Percussion marks are never systematically located on the same specific bone area in the different layers and sites. However, some tendencies emerge and the heterogeneous percussion mark distribution highlights some more marked areas than others do. These trends differ according to the elements and faunal assemblages. Our results show that the humeri and the femora of layer 4.1 of Abri du Maras present a non-random percussion mark distribution, which is different from that obtained in an intuitive context. The analyses demonstrate a non-random and counter-intuitive pattern for the percussion mark distribution of the assemblage 4.2LUNG humeri and the 4.2 MUNG tibias of Abri du Maras, and also of the Saint-Marcel Cave radio-ulnas. This distribution of percussion marks shows the probable existence of butchery traditions at Abri du Maras layers 4.1, 4.2, Saint-Marcel Cave layers g, and h. At Riparo Tagliente, this is not obvious, due perhaps to statistical biases. However, it was not possible to discriminate sites on the basis of percussion mark distribution, except for the metatarsals from Saint-Marcel and Abri du Maras, indicating a statistically significant difference between them.

### Taphonomic biases in the identification of percussion marks

In order to analyse further the distribution of percussion marks and their meaning, it is necessary to estimate the impact of the different taphonomic factors on faunal assemblages.

Taphonomic differences clearly exist between our sites. The percussion marks ratios from both layers at Abri du Maras are lower than at Riparo Tagliente and Saint Marcel Cave. This is partly due to the greater illegibility of the Abri du Maras bone surfaces (root-etching dissolution and desquamation). Indeed, the greatest difference between the assemblages is related to the ratio of pits. While the pits are the second most frequent traces at Saint Marcel Cave and Riparo Tagliente, at Abri du Maras, they are scarce or absent. Pits only alter the superficial cortical surface, whereas notches and adhering flakes penetrate the whole thickness and reach the medullary cavity [108, 131]. The degree of bone surface preservation may partly explain the limited number of pits in this case. Concretions may also sometimes partially or completely cover the surfaces. The same is true for abrasion alterations, such as polishing. However, oxide colouration, as well as fire surface alterations, do not seem to affect percussion marks.

The degree of post-depositional bone fragmentation should also be considered in the percussion marks analysis. In our assemblages, breakage was mainly perimortem, as evidenced by the high proportion of green bone fractures. However, some dry bone fractures were also

observed, sometimes related to green bone fractures. Post-depositional cracking, dry bone fractures, thermo-clastic can duplicate some percussion marks, or affect adhering flakes.

Concerning the impact of differential preservation on percussion mark distribution, we can correlate the absence of crushing marks in our assemblages with the lack of epiphyses. Whether this results from natural or anthropogenic differential preservation (bone fat recovery or fuel), we observed that shaft fragments are much more numerous than epiphyses. Yet, this type of mark is usually located close to the spongy portions [122].

To conclude, despite all these taphonomic biases, the percussion mark ratios (Number of remains with percussion marks / NISPa) recorded in our assemblages are still higher than in other sites, such as Fumane [132], Saint-Germain-la-Rivière [52] or Abric Romani [133]. This indicates systematic marrow extraction by percussion at the three compared sites.

## Percussion marks: A valuable indicator of butchery traditions?

None of our assemblages shows a systematic location of the percussion marks (Figs 4–11, 14). Our results seem to be more similar to those observed at Gran Dolina TD10-1 [43]. Percussion mark frequencies indicate some tendencies depending on the samples, but their location varies according to the elements or the assemblages, for example, the humerus (p3l) at Saint-Marcel Cave (layer g) and Abri du Maras (assemblage 4.2LUNG) (Fig 4). No correlation was observed between distribution and bone density or cortical thickness. However, percussion mark tendencies were not sufficient to evoke the presence of standardization. Areas with percussion marks also show tendencies rather than systematization. Before attempting to interpret them in terms of traditions, we would like to assess whether these observed tendencies are randomly caused, and if we can rule out the standardization of percussion mark distribution.

In our studied samples, we observe two cases: the first with random distribution of percussion marks, and the second with standardized distribution. In the first case, such as at Riparo Tagliente, we can suggest that several groups or individuals extracted marrow differently during recurrent occupations–palimpsest [60, 61, 106, 134]. Due to the limited number of remains at Riparo Tagliente, for example, both layers were analysed together, erasing possible differences. The high fragmentation of the bones to extract yellow and red marrow, in order to use the cancellous portions as fuel or to make tools, such as retouchers, could also erase the systematic percussion areas produced during yellow marrow extraction. Indeed, these activities could produce more percussion marks in different locations depending on the use of the bone fragment. The entire butchery 'chaine opératoire' of the long bone processing before and after the marrow recovery should be taken into account in each site and assemblage. Otherwise, respectively for the levels 35 and 37, 37.4% and 35.0% of the bones were burnt, almost all of them unidentified taxonomically. In both levels, a high number of retouchers was identified. Supplementary fractures and percussion marks could result from the bone being used as a tool.

The second case refers to bones where percussion marks are not randomly distributed, such as at Abri du Maras or Saint-Marcel Cave (tibias only) for all the levels analysed. Such cases suggest a yellow marrow extraction tradition or an intuitive way to break bones. We tested whether the identified standardization is similar to the intuitive pattern or not, in order to interpret the standardization as culturally induced. Some features of percussion mark distribution at Abri du Maras are non-random and are very different to the intuitive pattern (layer 4.1, humerus and femur; layer 4.2 of humerus of the LUNG assemblage and radio-ulna of the MUNG assemblage). This seems to imply probable butchery traditions for some elements, as suggested by our experimental data. Nevertheless, in layer 4.2, this non-random feature only concerns one bone of each sample, i.e., the LUNG (humerus) and the MUNG (radio-ulna)

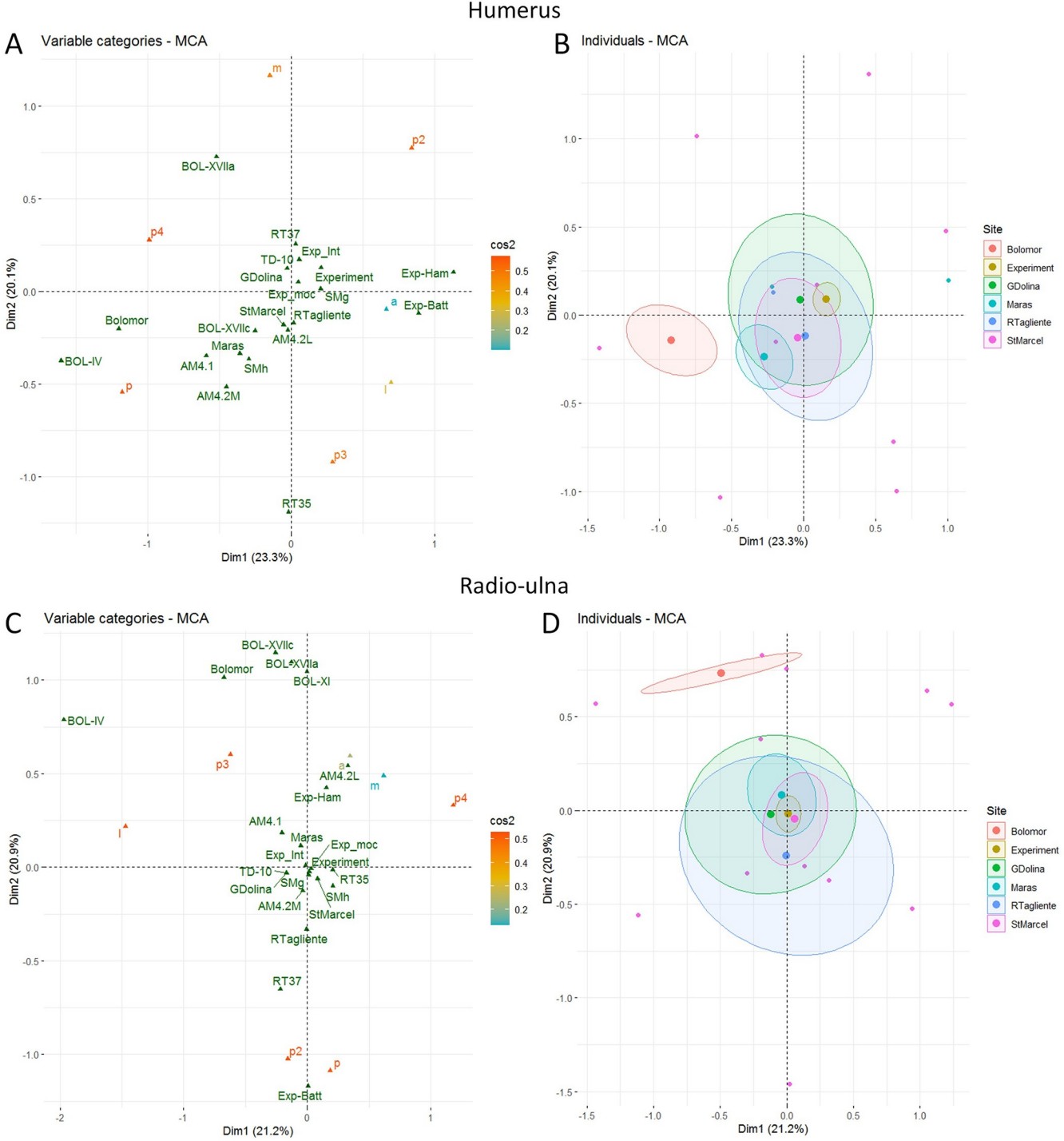

**Fig 14.** MCA of the areas with percussion marks by bone portion and side for humerus (A-B) and radio-ulnas (C-D). Sites are included as illustrative variable only. Cos$^2$ are displayed following a colour gradient.

assemblages. This difference could be explained by a dissimilar treatment of the two different-sized ungulates in the two layers. Data from the Abri du Maras assemblages demonstrate that one assemblage with non-random and counter-intuitive percussion mark distribution in one

level could influence a larger assemblage of one site. In other words, standardization could be highlighted in the occupation palimpsest. The association of two assemblages can be relevant, given the statistically significant results, as observed at Abri du Maras. However, standardization could only be highlighted by considering several layers, as at Saint Marcel.

In the case of the absence of randomization and counter-intuitive patterns, the hypothesis of a butchery tradition can be reasonably assumed when distribution is standardized. Specific and systematic yellow marrow extraction know-how could be partially erased by taphonomic processes or additional activities after marrow recovery. Furthermore, the marrow extraction process could necessitate breaking the diaphysis several times to recover the highest quantity of marrow [122, 135]. Consequently, it could be possible to identify standardized practices for some long bones, but not for all of them.

This methodological approach, which tests the standardized and counter-intuitive distribution with the chi-square test allows us to reveal possible butchery traditions in the assemblage of Abri du Maras layer 4.1, whereas the analysis of percussion mark frequencies failed to do so [29]. Together, both Saint-Marcel Cave layers demonstrate the existence of one tradition only for tibias. Groups with similar practices may have occupied the cave, regarding the tibia. They could have employed/used different practices on the other long bones.

At Abri du Maras, Neandertal groups with marrow recovery skills may have transmitted standardized practices and gestures. Furthermore, the practices of the group(s) in layer 4.1 seem to have been different from those of the group(s) in layer 4.2. This latter group processed bones differently, depending on the size of the ungulate. The absence of a clear systematization of percussion marks' location in the Abri du Maras assemblages could be explained in two ways: a complex "*chaîne opératoire*", or the occasional passage of groups with different butchery practices, or both.

However, this methodology is limited to the number of preserved areas with percussion marks. In our study, the emblematic example is the layers of Riparo Tagliente. Percussion mark frequencies indicate differences in bone treatment. However, due to the scarcity of remains, the chi-square analyses do not discern differences between layers 35 and 37. Thus, it is not possible to establish whether the lack of differences is due to the combined study of both layers nor to the absence of butchery traditions. The methodology needs to be refined to resolve such questions.

The distribution of the recorded percussion marks is never systematic at all the sites where percussion marks distribution was studied: in Abri du Maras, Saint-Marcel Cave, Riparo Tagliente, Gran Dolina [43], Fumane [132] and Saint-Germain-la-Rivière [52] with the exception of Bolomor (layer IV). In this specific layer of Bolomor, we observed one or two systematic areas with percussion marks. This could highlight a very original marrow extraction practice with specific marrow consumption, perhaps heated, that can be supported by the higher proportion of mixed angles [135].

## Interrelated traditions among technological and subsistence strategies

The hunting strategies of the three sites are different, as Neandertal groups selected different cervid species according to the sites. However, almost all the samples were practically mono-specific, except the 4.2LUNG (Abri du Maras). Lithic technologies varied from one site to another with changes between some layers. The comparison of these differences with possible butchery traditions enables us to conduct an in-depth investigation of the diversity of Neandertal behaviour at the intra or inter-site scale.

At Riparo Tagliente, lithic technologies change between layer 35 and 37 with the emergence of laminar core technology, whereas hunting strategies remained similar [18, 136, 137]. The

Neandertal groups occupying the site during the filling of these layers chose to hunt only roe deer, focusing on the ecological specialization of medium to large ungulates. It was not possible to identify specific standardized marrow recovery.

At the inter-site scale, the sites of Abri du Maras and Saint-Marcel are relatively close both geographically and chronologically (Fig 1). It is possible to propose the presence of the same groups with similar butchery traditions. However, the main core technology is Levallois/other technologies (cores on flakes or nodules, discoid-type cores) for Abri du Maras and discoid for Saint Marcel. Neandertal groups were highly mobile, and each assemblage can be due to complex occupation events depending on the season, animal migration or raw material requirements [138, 139]. Furthermore, each layer was an occupation palimpsest showing several successive occupations by one or several groups. Hunting strategies are characterized by an almost mono-specific spectrum of middle-sized ungulates for layer 4.1 of Abri du Maras and both layers of Saint-Marcel Cave. The site was occupied at a particular period of the year: autumn for Abri du Maras layer 4.1 and spring-summer for Saint-Marcel Cave and for layer 4.2 of Abri du Maras. However, yellow marrow is extracted differently at those two sites, in particular for metapodials. These results show that the two sites were occupied by different groups with no marrow recovery traditions at a regional scale, between Abri du Maras and Saint-Marcel Cave. Technical strategies were quite different in the two sites (Levallois and discoid), corroborating the hypothesis of groups with different technological and hunting and butchery strategies [37]. Besides, the study only concerned two sites in that region; it should be expanded to other regional sites. The contemporaneity of the sites is relative, since we are considering on large time scales. Nevertheless, these first results are promising, showing the importance of such a study for understanding the dynamics of occupation of Neandertal sites.

Regarding the intra site-layer comparison, the technical strategies are similar between both layers g and h of Saint-Marcel Cave. Moreover, hunting strategies are also similar. The red deer were the most hunted species in the two layers [80]. These common strategies devoted to one species could be reflected by similar butchery practices, as attested by the tibias and probably the metapodials.

At the Abri du Maras layers 4.1 and 4.2, technological behaviours differ slightly from one layer to the other (same diversity of core technologies but more retouched material for layer 4.2). We also observe a change of the prey choices regarding hunting strategies. Layer 4.1 is characterized by selective reindeer hunting [61]. We note standardized practices and possible specialization in marrow extraction regarding the stylopodials. The hunting and marrow recovery strategies are different in layer 4.2. Reindeer still dominates the faunal spectrum, but the proportions of other large ungulates are higher. We observe distinct marrow extraction methods for the different ungulate class-sizes for each assemblage. The breakage of the humerus is more standardized for large ungulates and the breakage of the radio-ulna is more standardized for middle-sized ungulates. This site illustrates slight changes in butchery traditions over time between two different phases of occupations. The groups occupying layer 4.2 extracted marrow differently depending on the size of the herbivores. Therefore, it was probable that two different groups with their own butchery traditions occupied the site over time.

Only some long bones from Abri du Maras and Saint-Marcel Cave were broken in a standard way. The standardly broken bones were different depending on the site or the assemblages. This bone treatment could point to task division or time-delayed consumption [27, 140, 141]. Based on ethnographic studies, we can propose several hypotheses to explain differential bone treatment. Several works focusing on current hunter-gatherer societies report that some bones were cracked directly on the kill-site for immediate marrow consumption ("snack"), while others were packed and transported to the base camp for delayed consumption; e.g. [27, 140–144]. Hunters seemed to prefer zeugopodia and metapodials for snacking

and raw consumption. In this case, the bone remains were abandoned on the site. The femora and humeri were transported to the base camp to be shared by the group. They were cooked and later broken to be eaten. Binford [118] observed task distribution for femora, which were broken by women at the camp base. Some groups, such as the Hadza, San, Maasai or Siberian groups as Evenks or Evens process carcasses differently, depending on the size of the hunted animals and/or depending on the skeletal element [141–143, 145–150]. The complexity of the dynamic of carcass transport and carcass processing needs to take into account, according to the animal size and element. If major trends can emerge, this dynamic could vary according to each carcass or situational event. These current hunter-gatherers groups use modern tools and adapt their behaviour to them, like container to cook or steel and iron axe or knife used during the carcasses process. We should be careful when making direct comparisons with these ethnographic examples regarding carcass processing and especially the fracture patterns. We could not apply this observation directly to our Neandertal group assemblages. However, the hypothesis of differential long bone treatment depending on the type of occupation could be backed up by the results from layer 4.1 of Abri du Maras. Only the stylopodia were traditionally broken to recover marrow, which could then have been cooked (see references above). The different treatment depending on species size observed in layer 4.2 between the 4.2MUNG and the 4.2LUNG was also observed in the current ethnographical groups (for example [143]) and also within archaeological assemblage of Qesem cave regarding the treatment of the metapodial [147].

Our data attest to variability in Neandertal behaviour during the Middle Palaeolithic, and not only for subsistence patterns. This variability can be partly explained by the geographical extension and duration of this period associated with the diversity of environmental conditions. Each context seems to represent a balanced response to specific conditions. Based on this assumption, the existence of different butchery practices is highly plausible. Variability in traditions could be noticed in subsistence, including both hunting and butchery strategies, and technical strategies, as observed in the sites and layers of Abri du Maras, Saint-Marcel Cave and Riparo Tagliente.

## Conclusion

Our results suggest the probable presence of butchery traditions in the Abri du Maras and Saint Marcel sites, particularly at Abri du Maras, where the level 4.1 assemblage presents standardized and counter-intuitive distributions of percussion marks on humeri and femora. The samples from level 4.2 show that tibias from medium-sized ungulates were processed traditionally (mainly reindeer), whereas for large ungulates, this was only the case for the humerus. This suggests a difference in marrow recovery processes related to animal size. For Saint-Marcel, the distribution of percussion marks on radio-ulnas shows standardized marrow extraction practices for the whole assemblages, but this standardization disappears when the levels are tested separately. This difference points to a relatively minor trend in both levels, which may not be distinguishable for a single level.

The distribution of percussion marks on the metapodials of the Abri du Maras and Saint-Marcel is not random. However, no comparative experimental sample is available for these elements, so it is not possible to determine if the marrow was extracted intuitively or counter-intuitively.

The groups that occupied the French sites processed some long bones in a standard way. However, the types of these long bones are different depending on the assemblages. This first analysis does not yield any information on the specificity of the patterns characterising these sites or levels, just to know that some share the same. It was not possible to put forward

hypotheses regarding Riparo Tagliente, in particular because of the limited number of bones studied. Through the comparison of all the sites, as well as intuitive experiments, based on the analyses of percussion marks distribution using exploratory analysis (MCA), we showed that for each element the Bolomor site almost always differs from the other sites and experiments. Those assemblages seem to be different from the other sites, showing specific/distinctive areas marked by percussion. The additional analysis of the Fisher Exact test did not reveal a site-specific percussion pattern based on a particular percussion zone. Nevertheless, we identified one exception: the Saint-Marcel Cave and Abri du Maras metatarsals, where the analyses revealed significant differences in the number of traces observed. For metatarsals, in these two sites, despite the absence of comparative experimental data, we can suggest the presence of traditional marrow extraction. For this element only, the traditions appear to be different from each other. For other sites and elements, it is not possible to discriminate between one site and another. These results suggest that despite relatively poor surface conservation, percussion marks are robust enough to demonstrate the existence of butchery traditions in anthropogenic accumulations during the Middle Palaeolithic. In this paper, we established how several taphonomic modifications could affect percussion traces in an archaeological context in Abri du Maras, Saint-Marcel and Riparo Tagliente. We suggest taking into consideration the degree of surface illegibility in percussion mark counts/estimation, especially for pits and grooves. Furthermore, differential preservation influences some traces, mostly crushing marks.

Such work based on the distribution of percussion marks is certainly time-consuming, but it enables us to further the debate on the socio-cultural practices of Neandertal groups. We need to pursue this research and increase the corpus of sites to enhance our understanding of the different standardized and counter-intuitive practices for the Middle and Upper Palaeolithic in order to discern differences between Neandertal and modern human groups. Beside, a current works has highlighted differences in thumb morphology implying that Neandertals have a better grip on voluminous objects than modern human [151]. These results could question the influence of the grip in the way to extract marrow with hammerstone, and how this could shape some traditional practices.

In the future, a new methodological approach based on spatial analysis to study percussion marks distribution should be tested in archaeological context [45]. This method takes into account the spatial relation of percussion traces and the preservation of long bone remains and facilitates analyses of the marks without the arbitrary division of long bones. This method, using GIS analyses, has provided results for intuitive experiments. The comparison between the intuitive pattern and the archaeological assemblages could fine-tune our comprehension of Neandertal and modern human butchery behaviours. It may also be necessary to carry out complementary experiments based on intuitive ways of recovering yellow marrow in particular on metapodials.

We can now consider the practice of bone marrow recovery as one of the structuring elements of a more global reflection on interactions and intergenerational transmission. These butchery traditions highlighted complex practices within human groups and practices that may diverged between layers of a site or show a certain continuity of butchering know-how. Our results interrogate if the traditions of lithic, hunting and butchery could be intertwined.

## Supporting information

**S1 File.**
(DOCX)

## Acknowledgments

Fieldworks at Abri du Maras were supported by the Regional Office of Archaeology in Rhône-Alpes, the French Ministry of Culture and Communication and the Ardèche Department through several scientific programs. L. Byrne, an official translator and native English speaker, edited the English manuscript. We are grateful to the reviewers and particularly to A. Val, for their very useful comments, which enhance a lot this manuscript.

## Author Contributions

**Conceptualization:** Delphine Vettese.

**Data curation:** Delphine Vettese, Ruth Blasco, Louis Chevillard, Trajanka Stavrova.

**Formal analysis:** Delphine Vettese, Antony Borel, Ruth Blasco, Louis Chevillard, Trajanka Stavrova.

**Funding acquisition:** Delphine Vettese, Ruth Blasco, Marie-Hélène Moncel, Camille Daujeard.

**Investigation:** Delphine Vettese, Antony Borel, Ursula Thun Hohenstein, Camille Daujeard.

**Methodology:** Delphine Vettese, Antony Borel, Ruth Blasco.

**Project administration:** Camille Daujeard.

**Resources:** Delphine Vettese.

**Supervision:** Delphine Vettese, Camille Daujeard.

**Validation:** Delphine Vettese, Antony Borel, Ruth Blasco, Ursula Thun Hohenstein, Marta Arzarello, Marie-Hélène Moncel.

**Visualization:** Delphine Vettese.

**Writing – original draft:** Delphine Vettese.

**Writing – review & editing:** Antony Borel, Ruth Blasco, Louis Chevillard, Trajanka Stavrova, Ursula Thun Hohenstein, Marta Arzarello, Marie-Hélène Moncel, Camille Daujeard.

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
