## [Decision Letter · Decision Letter 0]

13 Jan 2022

PONE-D-21-28076New evidence of Neandertal butchery traditions in Southwestern Europe (MIS 3-5)PLOS ONE

Dear Dr. Vettese,

Thank you for submitting your manuscript to PLOS ONE. After careful consideration, we feel that it has merit but does not fully meet PLOS ONE’s publication criteria as it currently stands. Therefore, we invite you to submit a revised version of the manuscript that addresses the points raised during the review process.

We look forward to receiving your revised manuscript.

Kind regards,

Enza Elena Spinapolice, Ph.D

Academic Editor

PLOS ONE

https://journals.plos.org/plosone/s/file?id=ba62/PLOSOne_formatting_sample_title_authors_affiliations.pdf”

2.  In your manuscript, please provide additional information regarding the specimens used in your study. Ensure that you have reported specimen numbers and complete repository information, including museum name and geographic location.

For more information on PLOS ONE's requirements for paleontology and archaeology research, see https://journals.plos.org/plosone/s/submission-guidelines#loc-paleontology-and-archaeology-research.

4. Thank you for stating the following in the Funding Section of your manuscript:

“CD, DV, TS and LC was supported by the Fondation Nestlé France (SJ 671–16) (https://fondation.nestle.fr/); DV is supported by the Centre d’Information des Viandes – Viande, sciences et société (SJ 334–17); RB is supported by a Ramón y Cajal research contract by the Ministry of Economy and Competitiveness (RYC2019-026386-I) and develops her work within the AEI/FEDER, EUproject PID2019-104949GB-I00, and the Generalitat de Catalunya projects 2017 SGR 836 and CLT009/18/00055; AB, CD, DV, LC, MHM and TS are supported by the Muséum national d’Histoire naturelle.

“RB:Ramón y Cajal research contract by the Ministry of Economy and Competitiveness (RYC2019-026386-I) and AEI/FEDER, EUproject PID2019-104949GB-I00, and the Generalitat de Catalunya projects 2017 SGR 836 and CLT009/18/00055; CD, DV, TS and LC was supported by the Fondation Nestlé France (SJ 671–16) (https://fondation.nestle.fr/); DV is supported by the Centre d’Information des Viandes – Viande, sciences et société (SJ 334–17); AB, CD, DV, LC, MHM and TS are supported by the Muséum national d’Histoire naturelle.

5. We note that [Figure 1] in your submission contain [map/satellite] images which may be copyrighted. All PLOS content is published under the Creative Commons Attribution License (CC BY 4.0), which means that the manuscript, images, and Supporting Information files will be freely available online, and any third party is permitted to access, download, copy, distribute, and use these materials in any way, even commercially, with proper attribution. For these reasons, we cannot publish previously copyrighted maps or satellite images created using proprietary data, such as Google software (Google Maps, Street View, and Earth). For more information, see our copyright guidelines: http://journals.plos.org/plosone/s/licenses-and-copyright.

 a. You may seek permission from the original copyright holder of [Figure 1]  to publish the content specifically under the CC BY 4.0 license. 

Reviewers' comments:

Reviewer's Responses to Questions

**Comments to the Author**

1. Is the manuscript technically sound, and do the data support the conclusions?

Reviewer #1: Yes

Reviewer #2: Yes

2. Has the statistical analysis been performed appropriately and rigorously? 

Reviewer #1: Yes

Reviewer #2: I Don't Know

3. Have the authors made all data underlying the findings in their manuscript fully available?

Reviewer #1: Yes

Reviewer #2: Yes

4. Is the manuscript presented in an intelligible fashion and written in standard English?

Reviewer #1: Yes

Reviewer #2: Yes

5. Review Comments to the Author

Reviewer #1: The manuscript PONE-D-21-28076 is a very interesting research on how Neandertals manage animal resources, revealing butchery traditions. The paper is focused on robust evidence from Southwestern Europe, although many colleagues with interest in other geographical locations and even chronologies would find this research very interesting. Overall, I find the paper very interesting, adequate for the journal, original both in the question and in the methods, and therefore I would recommend publication.

Specifically, all sections are well written, with up-to-date information. The introduction, for instance, nicely sets the state-of-the-art of the problematic, and frames correctly the question that is going to be addressed in the paper. Methods are correct, and discussion equilibrated.

The statistical treatment of the data is excellent and very useful.

Although, very minor review is required in my opinion:

· Figures are very nicely produced and they reflect the point that they are intending. They are supportive of the text, but I find that there are too many of them. There is an excessive use of figures that may me too many for the type of publications the authors aim to publish. Why not try to build a single schematic figure for Figures 3 to 8, 9 to 11 and 12 to 14. I realize this is a lot of work, and that it is not easy to condense such information in a single figure, but the use of so many figures would generate a very long paper with nice information being just roughly placed there (all that figures nicely produced by the authors could be supplementary information). For action: Please try and condense the sets of figures mentioned in single-basic figures.

· Figure 1 (map) requires a minor editing to make it look a bit better. It looks like a PPT slide, with sizes not being homogenized.

· P.36 (line 774). Please homogenize the use of Neandertal/Neanderthal. There may be others not spotted by me.

· Maras and Abri du Maras are used indistinctively. Why not use Abri du Maras in the first instance, and then use Maras? Or always use Abri du Maras. Please reconsider.

· Very minor English edits are required:

- P. 38 (line 826). The plural of femur is femora, not femurs. Please correct.

I would really like to see this manuscript published, and I hope the authors find these comments and reviews useful to produce a very interesting (and useful) paper for the scientific community.

Reviewer #2: I am impressed by the amount of work that clearly has gone into collecting and analyzing the data. I find the research question relevant and original. The results are interesting and will be relevant to a wide range of archaeologists interested in subsistence strategies, and not only in the Middle Palaeolithic.

The text would benefit from careful proof-reading and "clean-shaving". Having a native speaker translating a manuscript does not guarantee the quality of the writing, and there are many (many) simple grammatical mistakes, which should have been avoided. Besides, they are common scientific terms from the archaeo/zooarchaeo jargon that are misspelled, mistranslated or clumsy direct translations from French (e.g. indexes instead of indices; "series" instead of "assemblages" in English; the use of "bone marrow recovering" or terms like "traditions" and "traditional", which are sometimes inappropriately used, etc.). There is also a mix between American and British English throughout the text. The manuscript is long and compact with interesting and new data. In a way, the core of the work you are presenting is in the tables and figures (which are good and informative). The text is only just there to tell the readers what they are looking at when they see your tables and figures. You really want the readers to get your message effectively; delivering the information in the most concise and clear way is critical. At the moment, I find the manuscript long, wordy at times and not always straightforward. This will particularly be the case for non-French speakers, who might struggle to understand what you mean sometimes. I have corrected many small things directly into the PDF document but probably not everything. I am picky when it comes to the writing but there are hundreds of articles published every year in our discipline and it is impossible to keep up. Chances are that very few people will actually thoroughly read the manuscript from beginning to end so you want each sentence to be simple to understand.

The abstract could be sharper. You really want to extract the “juice” from your work for the reader and give a clear and straightforward summary of your work (which research question did you have, which methods did you use on which material to address this question) and the key results. I would remove any circumstantial information, not directly relevant. Things such as “regardless of climatic conditions” in the first sentence for instance is not necessary even though it might be true. I would add a sentence or two describing the methods that you have applied. I think that you should also propose a conclusion regarding the hypothesis of cultural learning amongst Neanderthals, which you pose in the second paragraph of your abstract - but it does not appear again elsewhere?

I agree that butchery practices have not been used to track cultural traditions, which are still largely defined by lithic industries. This is, however, one of the main goals that every zooarchaeologist has in mind when reconstructing butchery practices from a fossil assemblage. The way you have phrased it (“butchery techniques have rarely been evaluated until now to track traditions”) undermined this collective effort. It is a detail but I use it to illustrate how important the choice of words ca be – particularly in the abstract -.

I would probably develop on the selection of the layers and the sites: why these layers and not others? size of the samples? chronology? why only two? If time constraints, that is completely acceptable but then I would say so.

Presentation of the sites: you need to be clearer in the description of the stratigraphic units selected (chrono-cultural attribution, age when available) and you need to be consistent with the types of information you describe (e.g. microfauna is sometimes mentioned and sometimes not). For instance, all taxa are presented for the Italian site but not for Saint-Marcel.

One could discuss the choice of stratigraphically closed stratigraphic units for intra-site comparison - it might be more relevant to select more distinct layers?

I have made other comments directly into the PDF document.

6. PLOS authors have the option to publish the peer review history of their article (what does this mean?). If published, this will include your full peer review and any attached files.

Reviewer #1: No

Reviewer #2: **Yes: **Aurore Val

---

## [Author Response · Author response to Decision Letter 0]

10 Mar 2022

Date: Jan 13 2022 09:49AM

To: "Delphine Vettese" delphinevettese@aol.com

From: "PLOS ONE" plosone@plos.org

Subject: PLOS ONE Decision: Revision required [PONE-D-21-28076]

PONE-D-21-28076

New evidence of Neandertal butchery traditions in Southwestern Europe (MIS 3-5)

PLOS ONE

Dear Dr. Vettese,

Thank you for submitting your manuscript to PLOS ONE. After careful consideration, we feel that it has merit but does not fully meet PLOS ONE’s publication criteria as it currently stands. Therefore, we invite you to submit a revised version of the manuscript that addresses the points raised during the review process.

We look forward to receiving your revised manuscript.

Kind regards,

Enza Elena Spinapolice, Ph.D

Academic Editor

PLOS ONE

In addition: 

Dear Dr. Vettese,

We've checked your submission and before we can proceed, we need you to address the following issues:

1. In your manuscript, please provide additional information regarding the specimens used in your study. Ensure that you have reported specimen numbers and complete repository information, including museum name and geographic location.

For more information on PLOS ONE's requirements for paleontology and archaeology research, see https://journals.plos.org/plosone/s/submission-guidelines#loc-paleontology-and-archaeology-research.

We made these changes lines 260-264: “The archaeological material of the Saint Marcel Cave is preserved in the Cité de la Préhistoire of Orgnac (France), that of the Abri du Maras is temporary deposited in the Institut de Paléontologie Humaine of Paris (France) and will be transferred at in the Cité de la Préhistoire of Orgnac (France) and that of Riparo Tagliente is in the University of Ferrara (Italy). All necessary permits were obtained for the described study, which complied with all relevant regulations.” 

The exact number of specimen studied is stipulated in the manuscript lines 294, 303, 306 and 312. 

2. We note that [Figure 1] in your submission contain [map/satellite] images which may be copyrighted. All PLOS content is published under the Creative Commons Attribution License (CC BY 4.0), which means that the manuscript, images, and Supporting Information files will be freely available online, and any third party is permitted to access, download, copy, distribute, and use these materials in any way, even commercially, with proper attribution. For these reasons, we cannot publish previously copyrighted maps or satellite images created using proprietary data, such as Google software (Google Maps, Street View, and Earth). For more information, see our copyright guidelines: http://journals.plos.org/plosone/s/licenses-and-copyright.

a. You may seek permission from the original copyright holder of [Figure 1] to publish the content specifically under the CC BY 4.0 license. 

As suggested previously, we already change the map using: Natural Earth (public domain): http://www.naturalearthdata.com/ site. We add in the manuscript line 130: “The map was created using ArcGIS (ArcMap 10.4), and uses Natural Earth vector map data, https://www.naturalearthdata.com/downloads/).”

We've returned your manuscript to your account. Please resolve these issues and resubmit your manuscript within 21 days. If you need more time, please email the journal office at plosone@plos.org. We are happy to grant extensions of up to one month past this due date. If we do not hear from you within 21 days, we will withdraw your manuscript.

Please log on to PLOS Editorial Manager at https://www.editorialmanager.com/pone/ to access your manuscript. You will find your manuscript in the 'Submissions Sent Back to Author' link under the New Submissions menu. Be sure to remove your previous manuscript file if you are uploading a new file in response to these requests. After you've made the changes requested above, please be sure to view and approve the revised PDF after rebuilding the PDF to complete the resubmission process.

We are requesting these changes to comply with the PLOS ONE submission guidelines (https://journals.plos.org/plosone/s/submission-guidelines). Please note that we won't send your manuscript for review until you have resolved the above requests. 

Thank you for submitting your work to PLOS ONE and supporting our mission of Open Science.

Kind regards,

Edrian Nim Tolentino

PLOS ONE

https://journals.plos.org/plosone/s/file?id=ba62/PLOSOne_formatting_sample_title_authors_affiliations.pdf”

2. In your manuscript, please provide additional information regarding the specimens used in your study. Ensure that you have reported specimen numbers and complete repository information, including museum name and geographic location.

For more information on PLOS ONE's requirements for paleontology and archaeology research, see https://journals.plos.org/plosone/s/submission-guidelines#loc-paleontology-and-archaeology-research.

4. Thank you for stating the following in the Funding Section of your manuscript:

“CD, DV, TS and LC was supported by the Fondation Nestlé France (SJ 671–16) (https://fondation.nestle.fr/); DV is supported by the Centre d’Information des Viandes – Viande, sciences et société (SJ 334–17); RB is supported by a Ramón y Cajal research contract by the Ministry of Economy and Competitiveness (RYC2019-026386-I) and develops her work within the AEI/FEDER, EUproject PID2019-104949GB-I00, and the Generalitat de Catalunya projects 2017 SGR 836 and CLT009/18/00055; AB, CD, DV, LC, MHM and TS are supported by the Muséum national d’Histoire naturelle.

“RB:Ramón y Cajal research contract by the Ministry of Economy and Competitiveness (RYC2019-026386-I) and AEI/FEDER, EUproject PID2019-104949GB-I00, and the Generalitat de Catalunya projects 2017 SGR 836 and CLT009/18/00055; CD, DV, TS and LC was supported by the Fondation Nestlé France (SJ 671–16) (https://fondation.nestle.fr/); DV is supported by the Centre d’Information des Viandes – Viande, sciences et société (SJ 334–17); AB, CD, DV, LC, MHM and TS are supported by the Muséum national d’Histoire naturelle.

5. We note that [Figure 1] in your submission contain [map/satellite] images which may be copyrighted. All PLOS content is published under the Creative Commons Attribution License (CC BY 4.0), which means that the manuscript, images, and Supporting Information files will be freely available online, and any third party is permitted to access, download, copy, distribute, and use these materials in any way, even commercially, with proper attribution. For these reasons, we cannot publish previously copyrighted maps or satellite images created using proprietary data, such as Google software (Google Maps, Street View, and Earth). For more information, see our copyright guidelines: http://journals.plos.org/plosone/s/licenses-and-copyright.

 a. You may seek permission from the original copyright holder of [Figure 1] to publish the content specifically under the CC BY 4.0 license. 

We have changed the figure with open-source data. 

Reviewers' comments:

Reviewer's Responses to Questions

Comments to the Author

1. Is the manuscript technically sound, and do the data support the conclusions?

Reviewer #1: Yes

Reviewer #2: Yes

2. Has the statistical analysis been performed appropriately and rigorously? 

Reviewer #1: Yes

Reviewer #2: I Don't Know

3. Have the authors made all data underlying the findings in their manuscript fully available?

Reviewer #1: Yes

Reviewer #2: Yes

4. Is the manuscript presented in an intelligible fashion and written in standard English?

Reviewer #1: Yes

Reviewer #2: Yes

5. Review Comments to the Author

Reviewer #1: The manuscript PONE-D-21-28076 is a very interesting research on how Neandertals manage animal resources, revealing butchery traditions. The paper is focused on robust evidence from Southwestern Europe, although many colleagues with interest in other geographical locations and even chronologies would find this research very interesting. Overall, I find the paper very interesting, adequate for the journal, original both in the question and in the methods, and therefore I would recommend publication.

Specifically, all sections are well written, with up-to-date information. The introduction, for instance, nicely sets the state-of-the-art of the problematic, and frames correctly the question that is going to be addressed in the paper. Methods are correct, and discussion equilibrated.

The statistical treatment of the data is excellent and very useful.

Although, very minor review is required in my opinion:

· Figures are very nicely produced and they reflect the point that they are intending. They are supportive of the text, but I find that there are too many of them. There is an excessive use of figures that may me too many for the type of publications the authors aim to publish. Why not try to build a single schematic figure for Figures 3 to 8, 9 to 11 and 12 to 14. I realize this is a lot of work, and that it is not easy to condense such information in a single figure, but the use of so many figures would generate a very long paper with nice information being just roughly placed there (all that figures nicely produced by the authors could be supplementary information). For action: Please try and condense the sets of figures mentioned in single-basic figures.

Unfortunately, building a single schematic figure for the group, as suggested, will cause a loss of information.

The figures 3 to 8, if we group all the bones (or just the limb bones or fore bones together) the information about the percentages of percussion marks by area will be lost. 

For the histogram, we lose the part of the location (portion and side) which are essential. 

Finally, the MCAs are complex to read, even at this size (the site, caption…), so it seems to us difficult to group the figures without losing information.

· Figure 1 (map) requires a minor editing to make it look a bit better. It looks like a PPT slide, with sizes not being homogenized.

We have made the changes.

· P.36 (line 774). Please homogenize the use of Neandertal/Neanderthal. There may be others not spotted by me.

We have made the changes.

· Maras and Abri du Maras are used indistinctively. Why not use Abri du Maras in the first instance, and then use Maras? Or always use Abri du Maras. Please reconsider.

We have made the changes.

· Very minor English edits are required:

- P. 38 (line 826). The plural of femur is femora, not femurs. Please correct.

We have made the changes.

I would really like to see this manuscript published, and I hope the authors find these comments and reviews useful to produce a very interesting (and useful) paper for the scientific community.

We thank the reviewer for his comments. 

Reviewer #2: I am impressed by the amount of work that clearly has gone into collecting and analyzing the data. I find the research question relevant and original. The results are interesting and will be relevant to a wide range of archaeologists interested in subsistence strategies, and not only in the Middle Palaeolithic.

The text would benefit from careful proof-reading and "clean-shaving". Having a native speaker translating a manuscript does not guarantee the quality of the writing, and there are many (many) simple grammatical mistakes, which should have been avoided. Besides, they are common scientific terms from the archaeo/zooarchaeo jargon that are misspelled, mistranslated or clumsy direct translations from French (e.g. indexes instead of indices; "series" instead of "assemblages" in English; the use of "bone marrow recovering" or terms like "traditions" and "traditional", which are sometimes inappropriately used, etc.). There is also a mix between American and British English throughout the text. The manuscript is long and compact with interesting and new data. In a way, the core of the work you are presenting is in the tables and figures (which are good and informative). The text is only just there to tell the readers what they are looking at when they see your tables and figures. You really want the readers to get your message effectively; delivering the information in the most concise and clear way is critical. At the moment, I find the manuscript long, wordy at times and not always straightforward. This will particularly be the case for non-French speakers, who might struggle to understand what you mean sometimes. I have corrected many small things directly into the PDF document but probably not everything. I am picky when it comes to the writing but there are hundreds of articles published every year in our discipline and it is impossible to keep up. Chances are that very few people will actually thoroughly read the manuscript from beginning to end so you want each sentence to be simple to understand.

We have homogenized for UK English as suggested. We tried to simplify the text following the suggestions as we could. 

The abstract could be sharper. You really want to extract the “juice” from your work for the reader and give a clear and straightforward summary of your work (which research question did you have, which methods did you use on which material to address this question) and the key results. I would remove any circumstantial information, not directly relevant. Things such as “regardless of climatic conditions” in the first sentence for instance is not necessary even though it might be true. I would add a sentence or two describing the methods that you have applied. I think that you should also propose a conclusion regarding the hypothesis of cultural learning amongst Neanderthals, which you pose in the second paragraph of your abstract - but it does not appear again elsewhere?

In the abstract, we removed “cultural learning”. We add a sentence to describe the methods: “Statistical analyses as the chi-square test of independence were employed to verify if percussion mark locations were randomly distributed and if these distributions were different from the intuitive ones.” Lines 28-30.

I agree that butchery practices have not been used to track cultural traditions, which are still largely defined by lithic industries. This is, however, one of the main goals that every zooarchaeologist has in mind when reconstructing butchery practices from a fossil assemblage. The way you have phrased it (“butchery techniques have rarely been evaluated until now to track traditions”) undermined this collective effort. It is a detail but I use it to illustrate how important the choice of words can be – particularly in the abstract -.

We agree with reviewer, and modify this sentence according: “While lithic technology is largely used to define cultural patterns in human groups, despite dedicating research by zooarchaeologists, for now butchering techniques rarely allowed the identification of clear traditions, notably for ancient Palaeolithic periods.” And we add: “Long bone breakage for bone marrow recovering is a commonly observed practice in Middle Palaeolithic contexts, regardless of the climatic conditions. While lithic technology is largely used to define cultural patterns in human groups, despite dedicating research by zooarchaeologists, for now butchering techniques rarely allowed the identification of clear traditions, notably for ancient Palaeolithic periods.” Lines 16-20.

I would probably develop on the selection of the layers and the sites: why these layers and not others? size of the samples? chronology? why only two? If time constraints, that is completely acceptable but then I would say so.

The selected studied faunal assemblages are coming from relatively contemporaneous levels (MIS 4-3), and for two sites are situated closely in the same region, with one from a different region but relatively closed. As we said, we chose series with an MNE >100, that is why the level 36 of Tagliente was discarded. even if the other levels 35 and 37 are very anthropogenic with a high bone breakage intensity. Regarding the levels of Abri du Maras, levels 4.1 and 4.2 are in the same stratigraphic unit and separate from each other by a sterile level. Concerning the level 5, it was not already completely excavated at the time of the study. Regarding Saint Marcel Cave, the previous study and the related database allowed us only both levels g and h. 

Within the paper, we explain also our criteria: 

“Within a given class size category, only samples with an MNE greater than 100 are selected.”

“For the purpose of intra-site comparisons, we selected relatively stratigraphically close layers from each site, coming from the same unit. The layers chosen were highly anthropized, with butchery marks and limited modifications of carnivore activity.”

Presentation of the sites: you need to be clearer in the description of the stratigraphic units selected (chrono-cultural attribution, age when available) and you need to be consistent with the types of information you describe (e.g. microfauna is sometimes mentioned and sometimes not). For instance, all taxa are presented for the Italian site but not for Saint-Marcel. 

We added the faunal spectrum of the levels studied “The faunal spectrum of the levels g and h is composed, in order of abundance: Cervus elaphus, Capreolus capreolus, Capra caucasica, Dama sp., Sus scrofa, and Equus sp” Line 196-197.

One could discuss the choice of stratigraphically closed stratigraphic units for intra-site comparison - it might be more relevant to select more distinct layers?

We have made the changes. 

I have made other comments directly into the PDF document.

We have made the changes. 

We answer at some comments in the pdf below: 

These are two different things in my opinion. I would separate them. Using available resources within a given environments could include plants for instance and/or animal resources that don't need to be cooked. The ability to cook relates to the ability to make, maintain and use fire, which is a different skills. The references you are quoting refer to dietary composition and not to the use of fire.

We added some references related to cook part: 

10. Henry AG. Neanderthal Cooking and the Costs of Fire. 2017;58. doi:10.1086/692095

11. Morin E, Soulier M-C. New Criteria for the Archaeological Identification of Bone Grease Processing. Am Antiq. 2017;82: 96–122. doi:10.1017/aaq.2016.16

12. Hardy K, Buckley S, Collins MJ, Estalrrich A, Brothwell D, Copeland L, et al. Neanderthal medics? Evidence for food, cooking, and medicinal plants entrapped in dental calculus. Naturwissenschaften. 2012;99: 617–626. doi:10.1007/s00114-012-0942-0

13. Estalrrich A, El Zaatari S, Rosas A. Dietary reconstruction of the El Sidrón Neandertal familial group (Spain) in the context of other Neandertal and modern hunter-gatherer groups. A molar microwear texture analysis. J Hum Evol. 2017;104: 13–22. doi:10.1016/j.jhevol.2016.12.003

I find the use of "traditional" and "traditions" sometimes a little bit confusing. It is not always clear what you mean by that. If you are tracking the existing of butchery traditions in given sites and through time, then say that. The expression "traditional butchery practices" is confusing.

Within the text, we substituted for the two occurrences concerned “traditional butchery practices” by “butchery traditions”. 

Lines 60-61, we add to clarify our “through time, in a specific site and comparing sites with others”, as suggested. 

Following the suggestion, we have made this change: “In this paper, we describe standardized and counter-intuitive patterns of breaking bones. These patterns are consistent with butchery traditions shared by members and transmitted at other members of the same Neandertal groups.” Lines 67-68.

I'm not sure I would keep the "inter-generational". I would say that it is a given that all human groups (of Neandertals as well) were inter-generational but I'm not sure how your data demonstrates that the bone breakage was practiced by members of different age categories within the group.

It is important this term of inter-generational because it is in the main definition of tradition: transmission of knowledge from one generation to another. In our case, the levels studied are palimpsests of several occupations. If the same group comes to the site many times, we could propose the following hypotheses: it is members of different generations and, the group evolves according to the birth and death within the group. And the palimpsest imply repetitive occupations, and in our case of one group with the same tradition, it could be also the same group during many generations. These hypotheses could be supported by ichnology data from the Rozel site and the genetic data from El Sidron cave (Duveau et al. 2019 and Rosas et al. 2006 and 2013 and Lalueza-Fox et al. 2011). We mean, in that case, not a direct genealogic transmission, but of individuals who have different ages. 

We add in introduction: “The transmission of butchery knowledge from one generation to another is essential in our definition of tradition. Most of the Middle Palaeolithic sites and levels are palimpsest of several occupations. That means one or more groups could occupy the site. If we identify one butchery tradition within a level, we identify one group with members of different generations or over several generations, returned to the site multiple times. This hypothesis is based on other studies that have focused at the composition of Neanderthal group as the ichnology study or genetic research [49–51]. For example, the Neandertal group of le Rozel is composed by a majority of children and adolescents. This group is composed of a relatively reduced number of individuals, who repeatedly occupied a site and a layer, with different subsistence or technical strategies.” Lines 80-88.

Line 72, we substituted “apprehension” with “intuition”.

Line 72-73, we substituted “and other characteristics of the bone” by “as the morphology, the thickness of cortical bone, the tissues compacta or spongiosa” as suggested. 

Surely, there are some examples in the literature of cut marks being used to document standardized butchery practices? Maybe not so common in the Middle Palaeolithic but one example that comes to mind is the work of MC Soulier and E. Morin on meat storage (2016, in JHE):

"Cutmark data and their implications for the planning depth of Late Pleistocene societies"

Indeed, meat storage may involve standardized butchery practices, but this notion differs from the one of traditions, also implying repeatability. The question of the cutmarks could be, but for now, it is usually used to identify a butchery activity and not traditions specific at one site. 

Technically, the chaine operatoire includes acquisition as well - hunting clearly didn't take place at the site so to be accurate, you don't have the "complete" chaine op. on site.

In the paper, we specify the butchery “chaîne opératoire”. This exclude the hunting, which is not a butchery practice. 

You say that all assemblages are mono-specific, except unit 4.2 of Abri du Maras, but here it sounds like 4.1 is also not mono-specific? It's not clear.

Regarding the level 4.1, the faunal spectrum is composed by diverse species, but the reindeer largely dominate it, with almost 90% of the NISP. We add: “where reindeers largely dominate with almost 90% of the NISP” Lines 154-155.

We replace : “These results indicated that layer f of Saint-Marcel did indeed date to the late Mousterian.” by “Thus, the chronological attribution of Unit 7 seems to be rather MIS 3.” Lines 192-193.

Again, I find the description of the stratigraphy (for someone who isn't familiar with the site) confusing. Earlier, you mention "two main stratigraphic units" and now you are talking about Unit 7?

The Mousterian levels c to u were grouped into units, Debard, 1988.

Also, do you mean cobble or pebble? There are two different things from a geological point of view (cut-size is not the same).

In this paper, we used cobble as meaning stone hammer. It is a lithic terminology and not geology. 

I would start the discussion with this section. Acknowledge the possible biases but start with those, otherwise, you leave the reader with a negative impression about your results. Let's look at the bright side - it is important to highlight what might have affected your assemblages, but you should conclude this section on a positive note and the large size of remains that could still be analysed and which exciting results you got from them.

We made this choice to present first the main results about the Neandertal butchery traditions and after how to demonstrate it. 

Is the high frequency of burnt bone a strong enough argument to posit that bone was used as fuel? 

I am more familiar with South African sites, where there are often high % of burnt bones but no clear evidence at all for the use of bone as fuel. Check Clark & Ligouis 2010 on this topic for instance.

Clark, J.L. & Ligouis, B. 2010. Burned bone in the Howieson’s Poort and post-Howieson’s Poort Middle Stone Age deposits at Sibudu (South Africa): behavioral and taphonomic 

My understanding is that you need to combine several lines of evidence to demonstrate that bone was used as fuel, so unless that work has been published somewhere, I would be more careful here.

We removed the sentence. It will be publish it in a future work. 

What do you mean here? Lithic technologies? Prey acquisition techniques? I find it confusing.

We changed “technical behaviour” by “lithic technologies” lines 807 and 811 as suggested. 

This is a good example of how the writing should be simplified and how confusing it can be. I don't know what you mean here and you will tire the reader if she/he has to try to understand which idea you are trying to convey.

We changed “Nevertheless, the complexity of dynamic of carcass transport and therefore, carcass processing proves to be of an obvious degree of difficulty.” by “The complexity of the dynamic of carcass transport and carcass processing needs to take into account, according to the animal size and element.” lines 863-865.

I find that often "complex" is not informative. Be straightforward, either you can compare modern with archaeological data or you can't.

We changed “It is complex to compare directly the carcass processes” by “We should be careful when making direct comparisons with these ethnographic examples regarding the carcass processing.” Lines 831-832.

6. PLOS authors have the option to publish the peer review history of their article (what does this mean?). If published, this will include your full peer review and any attached files.

Do you want your identity to be public for this peer review? For information about this choice, including consent withdrawal, please see our Privacy Policy.

Reviewer #1: No

Reviewer #2: Yes: Aurore Val

---

## [Decision Letter · Decision Letter 1]

8 Jul 2022

New evidence of Neandertal butchery traditions through the marrow extraction in southwestern Europe (MIS 5-3)

PONE-D-21-28076R1

Dear Dr. Vettese,

We’re pleased to inform you that your manuscript has been judged scientifically suitable for publication and will be formally accepted for publication once it meets all outstanding technical requirements.

Kind regards,

Enza Elena Spinapolice, Ph.D

Academic Editor

PLOS ONE

Reviewers' comments:

Reviewer's Responses to Questions

**Comments to the Author**

1. If the authors have adequately addressed your comments raised in a previous round of review and you feel that this manuscript is now acceptable for publication, you may indicate that here to bypass the “Comments to the Author” section, enter your conflict of interest statement in the “Confidential to Editor” section, and submit your "Accept" recommendation.

Reviewer #2: All comments have been addressed

2. Is the manuscript technically sound, and do the data support the conclusions?

Reviewer #2: Yes

3. Has the statistical analysis been performed appropriately and rigorously? 

Reviewer #2: Yes

4. Have the authors made all data underlying the findings in their manuscript fully available?

Reviewer #2: Yes

5. Is the manuscript presented in an intelligible fashion and written in standard English?

Reviewer #2: Yes

6. Review Comments to the Author

Reviewer #2: The authors have done a very good job at going through the various comments and suggestions and I applaud them for conducting a thorough revision. I'm happy with the revised version.

I have one final comment regarding the expression "marrow recovery", which the authors sometimes use. The common expression is “marrow extraction”, not “marrow recovery” (check the literature – the Anglo-Saxon literature; in fact, because I wasn't 100% sure myself, I did check this point with an American zooarch, J. Speth, and he confirmed that it is better to use marrow extraction). They mean slightly different things and as it is a key aspect of this study, I think that it is important to use the correct expression.

I have made final, minor edits directly into the PDF document.

Great work otherwise.

7. PLOS authors have the option to publish the peer review history of their article (what does this mean?). If published, this will include your full peer review and any attached files.

Reviewer #2: **Yes: **Aurore Val

---

## [Editor Report · Acceptance letter]

15 Jul 2022

PONE-D-21-28076R1 

New evidence of Neandertal butchery traditions through the marrow extraction in southwestern Europe (MIS 5-3) 

Dear Dr. Vettese:

I'm pleased to inform you that your manuscript has been deemed suitable for publication in PLOS ONE. Congratulations! Your manuscript is now with our production department. 

Kind regards, 

on behalf of

Dr. Enza Elena Spinapolice 

Academic Editor

PLOS ONE